# PADD: Path-Aligned Decompression Distillation for Non-Router Teacher to Guide MoE Student Learning

**Xinyue Peng** [1]   **Yi Qian** [1]   **Jiaojiao Lin** [1]   **Wenjian Shao** [1]   **Yanming Liu** [2]

## Abstract

As large language models (LLMs) continue to scale, it becomes increasingly challenging to grow model capacity under fixed computation budgets. We propose Path-Aligned Decompression Distillation (PADD), a framework for distilling knowledge from dense teachers without explicit routing into mixture-of-experts (MoE) students while learning high-quality routing policies. PADD organizes knowledge distillation into four stages in two phases: an initialization phase (Stage I) that builds diverse functionality in the student's experts through teacher neuron clustering and student-expert warmup, and a training phase (Stages II–IV) that integrates online adaptive distillation, path-refined policy optimization, and reward-augmented load balancing in a single training pipeline. Experiments on mathematical reasoning benchmarks demonstrate that PADD yields substantial gains over strong baselines at the same inference cost and that the MoE student can match or surpass its dense teacher. They also demonstrate effective teacher-to-student knowledge distillation and stable routing behavior.

## 1. Introduction

As large language models (LLMs) scale, the tension between model capacity and limited compute budgets grows. Dense models face bottlenecks in training throughput, inference latency, and memory bandwidth when scaling to hundreds of billions or trillions of parameters (Kaplan et al., 2020; Hoffmann et al., 2022). Mixture-of-Experts (MoE) architectures decouple parameters into sparsely activated expert subnetworks, decoupling capacity from inference FLOPs (Shazeer et al., 2017; Fedus et al., 2022). MoE can decompress entangled dense representations into structured expert modules (Komatsuzaki et al., 2023; DeepSeek-AI et al., 2024). However, most high-performing models remain dense; training MoE from scratch is expensive, and MoE-to-MoE distillation lacks generality due to incompatible expert decompositions and routing strategies (Dai et al., 2022; Zhang et al., 2025). In addition, using dense teachers offers flexibility: one can choose the best teacher per domain to provide task-specialized knowledge to a MoE student without increasing inference cost.

Converting dense to MoE faces a fundamental challenge: MoE relies on routing decisions, but dense models lack explicit routers. While one can perform sparse upcycling to match parameter shapes (Komatsuzaki et al., 2023), the new router has no supervision from dense teacher activations and must learn from scratch. This leads to router cold start (Dai et al., 2022): early training cannot distinguish syntactic from reasoning tokens, causing random noise diffusion across experts (logic diffusion). Conventional distillation only aligns outputs (Hinton et al., 2015) and cannot transmit internal processing preferences; discrete routing jumps break chain-of-thought continuity, causing path rupture (Zoph et al., 2022) and destabilizing gradients. When the MoE student's capacity (e.g., active parameters per token) is much smaller than the dense teacher's, a severe capacity gap prevents absorption of fine-grained logits (Gu et al., 2024).

However, static logits alignment is insufficient, and dynamic feedback is needed to guide routing. Reinforcement learning (RL) can in principle provide such feedback via rewards, and on-policy distillation (e.g., GRPO (Shao et al., 2024)) couples distillation with policy optimization so that teachers supervise students along their actual trajectories. Yet existing methods largely target dense-to-dense or MoE-to-MoE distillation and cannot bridge the structural mismatch between dense and MoE models. Classical MoE load balancing (Shazeer et al., 2017) only controls activation frequency, ignoring expert quality and leading to expert homogenization, while methods such as StableMoE, RSPO, and R3 (Dai et al., 2022; Zhang et al., 2025; Ma et al., 2025) stabilize routing for already-trained MoE models but assume usable expert structures and cannot recover path-level semantics from dense teachers. As a result, they fail to address the

---

[1]Intel Corporation, China [2]Zhejiang University, China. Correspondence to: Xinyue Peng <bearrr310@outlook.com>, Yi Qian <yi.qian@intel.com>.

*Proceedings of the 43rd International Conference on Machine Learning*, Seoul, South Korea. PMLR 306, 2026. Copyright 2026 by the author(s).

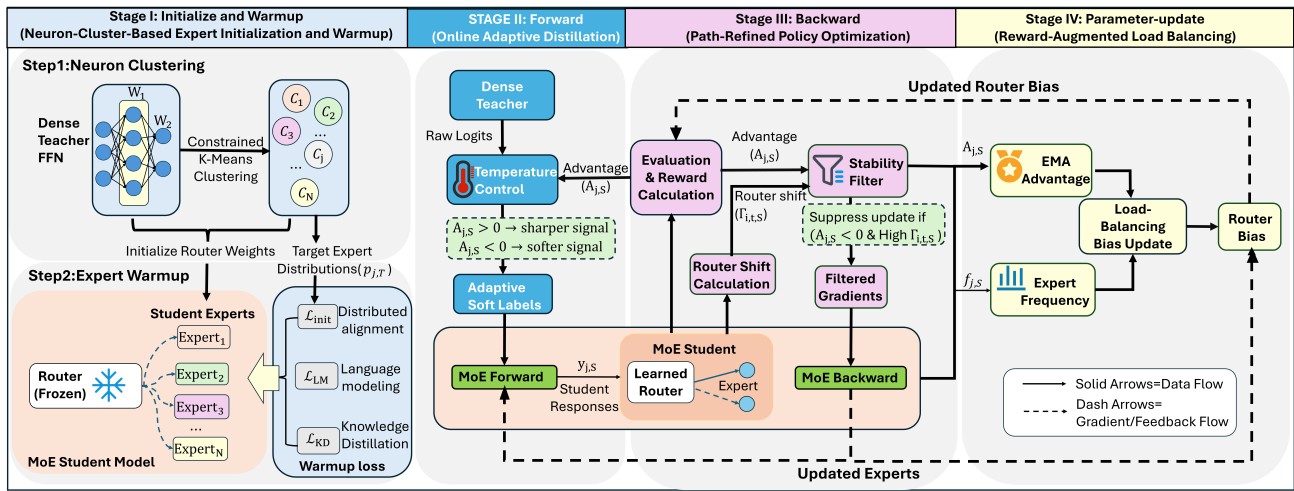

*Figure 1.* Path-Aligned Decompression Distillation (PADD) framework overview. PADD organizes dense-to-MoE knowledge distillation into four stages: Stage I performs neuron clustering and expert initialization, while Stages II–IV integrate online adaptive distillation, path-refined policy optimization, and reward-augmented load balancing in a unified training pipeline.

core challenge: structural deficits arising from architectural mismatch.

PADD targets a complementary setting to sparse upcycling (Komatsuzaki et al., 2023) and MoEfication-style conversion (Zhang et al., 2022): we transfer knowledge and routing from a router-less dense teacher into an already pre-trained router-aware MoE student, rather than constructing a new MoE from a dense checkpoint. The goal is to recover the teacher's implicit modular structure and to learn stable student routing under a fixed inference budget, while upcycling-style pipelines first build expert structure from dense weights and then train routing largely from scratch.

Figure 1 summarizes our proposed Path-Aligned Decompression Distillation (PADD) framework. PADD organizes distillation into four stages across two phases: an initialization phase (Stage I) that clusters teacher FFN neurons to initialize student experts and warms up experts into distinct functional roles; and a training phase (Stages II–IV) that combines online adaptive distillation, path-refined policy optimization, and reward-augmented load balancing in one pipeline. Together, these stages recover learnable path structure from the dense teacher and stabilize expert specialization at fixed inference cost.

Our main contributions are summarized as follows:

- We propose Path-Aligned Decompression Distillation (PADD), a unified framework for dense-to-MoE distillation organized into four stages across two phases: an initialization phase (Stage I) for expert setup and a training phase (Stages II–IV) that integrates online adaptive distillation, path-refined policy optimization, and reward-augmented load balancing. PADD system-

atically addresses router cold start, capacity gap, path rupture, and expert homogenization, enabling MoE students to match or even surpass dense teachers under the same inference cost.

- By analyzing neuron activation patterns in the dense teacher's FFNs and performing clustering, we construct student expert initialization and use warmup training to endow student experts with distinct functional roles corresponding to different neuron groups in the teacher, thereby mitigating router cold start and expert homogeneity at the source.

- We propose an online adaptive distillation mechanism that delivers smooth and absorbable semantic supervision along the student's actual routing paths, and introduce Path-Refined Group Relative Policy Optimization (PR-GRPO), which leverages routing shifts to suppress gradient instability caused by discrete routing and significantly improves the stability of MoE reinforcement learning.

- We propose reward-augmented load balancing, which jointly models expert activation frequency and performance quality, increases the activation probability of high-quality experts, alleviates the "expert homogenization" phenomenon induced by traditional load balancing, and enhances the long-term division of labor among MoE experts.

## 2. Methodology

In dense-to-MoE distillation, dense teachers lack explicit routing, so conventional offline distillation cannot guide MoE students to learn routing decisions. PADD combines

knowledge distillation with reinforcement learning through four stages organized into two phases: an initialization phase (Stage I) that performs initialization and warmup; and a training phase (Stages II–IV) that integrates online adaptive distillation, path-refined policy optimization, and reward-augmented load balancing in a unified training pipeline. We partition dataset $\mathcal{D}$ into four non-overlapping subsets: $\mathcal{D}_A$ for activation statistics and clustering in Stage I, $\mathcal{D}_B$ for expert warmup in Stage I, $\mathcal{D}_C$ for main training in Stages II–IV, and $\mathcal{D}_D$ for evaluation. We use subscripts T and S to distinguish teacher and student quantities (e.g., $p_\mathrm{T}$, $A_{i,\mathrm{S}}$). Before starting the PADD multi-stage training of dense-to-MoE distillation, we apply standard GRPO to the pretrained dense teacher to learn task-specific reasoning strategies that can be effectively distilled to MoE students.

## 2.1. Stage I: Neuron-Cluster-Based Expert Initialization and Warmup Alignment

Stage I has two steps using non-overlapping datasets $\mathcal{D}_A$ and $\mathcal{D}_B$. Expert warmup uses only $\mathcal{D}_B$ and does not reuse samples from $\mathcal{D}_A$, so that clustering statistics are not leaked into expert fitting. The first step performs activation statistics and neuron clustering on the teacher to construct target functional structures for student experts. The second step performs warmup alignment on student experts with frozen routers, forming initial functional differences aligned with the teacher's implicit structure.

**Activation statistics and clustering.** Let $L_\mathrm{T}$ and $L_\mathrm{S}$ be the number of layers in the teacher and student. The teacher is dense; each layer's feed-forward network (FFN) has two linear layers: $W_1 \in \mathbb{R}^{d_{ff,\mathrm{T}} \times d}$ (where $d$ is the hidden dimension) and $W_2$, with an activation function between them. Student layer $l$ corresponds to teacher layer $\lfloor l \cdot L_\mathrm{T}/L_\mathrm{S} \rfloor$. This holds no matter which has more layers, giving a unique teacher layer per student layer. The student is MoE with $N$ experts per layer; each expert is a 2-layer FFN. We extract the teacher FFN's $W_1$, where row $k$ ($w_k \in \mathbb{R}^d$) is the $k$-th neuron's weight vector. When $d_{ff,\mathrm{T}} \neq d_{ff,\mathrm{S}}$, we adjust teacher neurons to $d_{ff,\mathrm{S}}$ via uniform sampling, denoting the adjusted dimension as $d_{ff}$. Dense models lack explicit routing, but their FFN neurons co-activate under similar inputs, revealing implicit modular structure (Qiu et al., 2024). We perform cardinality-constrained K-Means clustering on $w_k$ to partition teacher neurons into $N$ clusters $C_j$, each corresponding to student expert $E_{j,\mathrm{S}}$:

$$\min_C \sum_{j=1}^N \sum_{k \in C_j} \|w_k - \mu_j\|^2, \quad \text{s.t. } |C_j| = \frac{d_{ff}}{N} \quad (1)$$

where $\mu_j \in \mathbb{R}^d$ is the centroid of cluster $C_j$. After obtaining $C_j$, we compute the activation distribution $p_{j,\mathrm{T}}$ of cluster $C_j$ on dataset $\mathcal{D}_A$, which will then be the learning target for student expert $j$. More specifically, for each sample $x \in$

$\mathcal{D}_A$, we forward it to the corresponding teacher FFN layer and record the first linear layer output (FFN intermediate activation) $h_k^{(x)} \in \mathbb{R}$ for neuron $k$. We average neurons in cluster $C_j$ over all samples in $\mathcal{D}_A$ to get $\bar{h}_j$, then apply softmax over $j = 1, \ldots, N$ to obtain $p_{j,\mathrm{T}}$:

$$\bar{h}_j = \frac{1}{|\mathcal{D}_A|} \sum_{x \in \mathcal{D}_A} \frac{1}{|C_j|} \sum_{k \in C_j} h_k^{(x)},$$
$$p_{j,\mathrm{T}} = \frac{\exp(\bar{h}_j/\xi)}{\sum_{j'=1}^N \exp(\bar{h}_{j'}/\xi)}, \quad \xi > 0, \quad (2)$$

where $\xi$ is the temperature parameter controlling the softmax sharpness. The distribution $p_{j,\mathrm{T}}$ guides student expert learning in the second step via $\mathcal{L}_\mathrm{init}$. This step only performs statistics and does not train the student.

**Expert warmup.** We initialize student router weights by mapping cluster centroids $\mu_j$ to the router's Linear weights at each layer. We then perform student expert warmup on $\mathcal{D}_B$ with frozen routers to prevent early routing instability and ensure all experts receive equal training signals before learning functional differences. Routing is fixed to a uniform distribution (each expert activated with probability $1/N$), so we train only the expert networks. Let $(x, y)$ be an input–output pair, $t$ be the token position, $\pi_\mathrm{S}$ be the student policy, and $p_\mathrm{T}$ and $p_\mathrm{S}$ be the teacher and student next-token distributions given context. The warmup loss includes language modeling loss $\mathcal{L}_\mathrm{LM} = -\mathbb{E}_{(x,y)\sim\mathcal{D}_B} \sum_{t=1}^{|y|} \log \pi_\mathrm{S}(y_t|x, y_{<t})$ and knowledge distillation loss $\mathcal{L}_\mathrm{KD} = \mathbb{E}_{(x,y)\sim\mathcal{D}_B} \sum_{t=1}^{|y|} D_\mathrm{KL}\big(p_\mathrm{T}(\cdot|x, y_{<t}) \,\|\, p_\mathrm{S}(\cdot|x, y_{<t})\big)$. We also introduce target activation distribution alignment loss $\mathcal{L}_\mathrm{init}$ to guide experts to learn functional differences aligned with the teacher's implicit structure:

$$\mathcal{L}_\mathrm{init} = \sum_{j=1}^N \mathrm{KL}(p_{j,\mathrm{S}} \| p_{j,\mathrm{T}}) \quad (3)$$

where $p_{j,\mathrm{S}}$ is the activation distribution of student expert $E_{j,\mathrm{S}}$ under the current input. The total warmup loss is:

$$\mathcal{L}_\mathrm{warmup} = \mathcal{L}_\mathrm{LM} + \alpha \mathcal{L}_\mathrm{KD} + \beta \mathcal{L}_\mathrm{init} \quad (4)$$

where $\alpha$ and $\beta$ are the distillation and target distribution weights.

## 2.2. Stage II: Online Adaptive Distillation

Stages II–IV execute sequentially in a single training run on $\mathcal{D}_C$: at each step, we sample $x \sim \mathcal{D}_C$, perform Stage II in the forward pass, Stage III in the backward pass, and Stage IV during parameter updates.

To address the capacity gap, we introduce online adaptive distillation in the forward pass. The GRPO-trained teacher

outputs $\text{Logits}_\text{T}$ containing semantic and task strategy information. This mechanism adaptively adjusts teacher log-probabilities based on student performance along the current expert path. Specifically, for group size $G$ and input $x$, we sample $G$ student responses $y_1, \ldots, y_G$. Let $r(x, y_i)$ be the reward for $y_i$, and $\bar{r}$ and $\sigma_r$ be the mean and standard deviation of group rewards. The student's within-group relative advantage is $A_{i,\text{S}} = (r(x, y_i) - \bar{r})/\sigma_r$. Since the teacher has no explicit router, we adjust teacher log-probabilities based on $A_{i,\text{S}}$'s assessment of the student path. The adjusted teacher next-token distribution is:

$$p_\text{T}^*(y|x) = \text{Softmax}\left(\frac{\text{Logits}_\text{T}}{\tau \cdot \Phi(A_{i,\text{S}})}\right) \quad (5)$$

where $\text{Logits}_\text{T}$ are the teacher's original logits, $\tau$ is the initial temperature, and the adjustment factor is:

$$\Phi(A_{i,\text{S}}) = 1 + \tanh(\kappa\, A_{i,\text{S}}), \quad (6)$$

where $\kappa$ is the response coefficient. Through advantage–temperature coupling, when $A_{i,\text{S}} > 0$, supervision strengthens (lower temperature, more confident signals); when $A_{i,\text{S}} < 0$, bias is corrected (higher temperature, more exploration). This prevents overfitting to incorrect expert paths.

### 2.3. Stage III: PR-GRPO Path-Refined Policy Optimization

In the backward pass, to mitigate policy gradient instability from routing shifts, we introduce Path-Refined Group Relative Policy Optimization (PR-GRPO). The student has $L_\text{S}$ MoE layers, each with one router and $N$ experts. The router is a single linear layer mapping hidden state $h \in \mathbb{R}^d$ to $N$-dimensional logits; after Softmax this yields $G_{\theta,\text{S}}(x_t) \in \mathbb{R}^N$. Each layer routes independently via Top-$K$ selection, forwarding through the corresponding 2-layer FFN. For multiple layers, we aggregate routing outputs before computing shifts.

Let $\pi_{\theta,\text{S}}$ be the student policy, $\theta_\text{old}$ be parameters from the previous update, and $x_t$ be the input at time $t$. The routing shift $\Gamma_{i,t,\text{S}}$ measures router decision changes relative to the previous step:

$$\Gamma_{i,t,\text{S}} = \|G_{\theta,\text{S}}(x_t) - G_{\theta_\text{old},\text{S}}(x_t)\|_2 \quad (7)$$

Let $a_t$ be the token action and $s_t$ be the state. We define the adjusted importance ratio as:

$$\hat{\rho}_t(\theta) = \frac{\pi_{\theta,\text{S}}(a_t|s_t)}{\pi_{\theta_\text{old},\text{S}}(a_t|s_t)} \cdot \exp(-\lambda \cdot \Gamma_{i,t,\text{S}} \cdot \mathbb{I}(A_{i,\text{S}} < 0)) \quad (8)$$

where $\lambda$ is the suppression coefficient, $\mathbb{I}(\cdot)$ is the indicator function, and $A_{i,\text{S}} < 0$ indicates poor path performance. When $A_{i,\text{S}} < 0$ and $\Gamma_{i,t,\text{S}}$ is large, the exponential term reduces the importance ratio, suppressing updates on unstable

paths. The PR-GRPO objective is:

$$\mathcal{J}_\text{PR-GRPO}(\theta) = \mathbb{E}_{x\sim\mathcal{D},\{y_i\}_{i=1}^G \sim \pi_{\theta_\text{old}}(\cdot|x)} \left[\frac{1}{G}\sum_{i=1}^{G}\frac{1}{|y_i|}\sum_{t=1}^{|y_i|}\right.$$

$$\min\left(\frac{\pi_\theta(y_{i,t}|x, y_{i,<t})}{\pi_{\theta_\text{old}}(y_{i,t}|x, y_{i,<t})} \cdot \exp(-\lambda \cdot \Gamma_{i,t} \cdot \mathbb{I}(A_i < 0)) \cdot A_i,\right.$$

$$\text{clip}\left(\frac{\pi_\theta(y_{i,t}|x, y_{i,<t})}{\pi_{\theta_\text{old}}(y_{i,t}|x, y_{i,<t})} \cdot \exp(-\lambda \cdot \Gamma_{i,t} \cdot \mathbb{I}(A_i < 0)),\right.$$

$$\left.\left.1 - \epsilon, 1 + \epsilon\right) \cdot A_i\right) \right] \quad (9)$$

where $G$ is the group size, $\epsilon$ is the clipping range, and $\hat{\rho}_{i,t}(\theta)$ at the token level follows Equation (8) with $a_t|s_t$ replaced by $y_{i,t}|x, y_{i,<t}$. The expectation is over $\mathcal{D}_C$. PR-GRPO adjusts importance ratios via routing shifts $\Gamma_{i,t,\text{S}}$, reducing weights on unstable samples and stabilizing gradients.

### 2.4. Stage IV: Reward-Augmented Load Balancing

During parameter updates, to address load balancing's neglect of expert quality, we apply reward-augmented load balancing to routing biases. For expert $j$, we track activation frequency $f_{j,\text{S}}$ (proportion of tokens routed to this expert) and within-group relative advantage $A_{j,\text{S}}$ (advantage of tokens routed to this expert). Since $A_{j,\text{S}}$ fluctuates during training, we smooth it with exponential moving average:

$$\text{EMA}(A_{j,\text{S}})_u = \lambda_\text{ema}A_{j,\text{S}} + (1 - \lambda_\text{ema})\text{EMA}(A_{j,\text{S}})_{u-1} \quad (10)$$

where $\lambda_\text{ema}$ is the decay coefficient, $u$ is the update step, and $\text{EMA}(A_{j,\text{S}})_0 = 0$. Ideally, $f_{j,\text{S}} \approx \bar{f} = 1/N$, while high-performing experts receive more activations. At each update cycle, we update routing bias $b_{j,\text{S}}$ for expert $j$:

$$b_{j,\text{S}}^{(\text{new})} = b_{j,\text{S}}^{(\text{old})} + \eta(f_{j,\text{S}} - \bar{f}) + \gamma \cdot \text{EMA}(A_{j,\text{S}})_u \quad (11)$$

where $\eta$ and $\gamma$ are traffic balance and reward compensation coefficients. Bias $b_{j,\text{S}}$ is added to the router's Linear layer logits at each layer before Softmax, adjusting expert $j$'s selection probability during Top-$K$ selection. The first term balances traffic; the second term augments rewards. High-quality experts receive positive bias increments and are prioritized, while the first term maintains balanced load.

## 3. Experiments

### 3.1. Experimental Setup

**Model selection.** We use two teacher–student pairs: **(1) Qwen family**: Qwen2.5-Math-7B (Yang et al., 2024) with Qwen3-30B-A3B (Yang et al., 2025); **(2) DeepSeek family**: DeepSeek-Math-7B (Shao et al., 2024) with DeepSeek-V2-Lite (DeepSeek-AI et al., 2024). Teachers are dense 7B models. Students are pretrained MoE checkpoints with

*Table 1.* Performance comparison of student models (MoE) and baseline methods on two families. The "Teacher (GRPO)" row shows teacher performance after GRPO on five benchmarks as a reference upper bound.

| Method | Qwen Family | | | | | | DeepSeek Family | | | | | |
|---|---|---|---|---|---|---|---|---|---|---|---|---|
| | AIME24 | AMC23 | MATH500 | Minerva | Olymp. | Avg | AIME24 | AMC23 | MATH500 | Minerva | Olymp. | Avg |
| Teacher (GRPO) | 83.0 | 94.7 | 91.3 | 55.5 | 63.8 | 77.7 | 55.3 | 69.2 | 79.5 | 36.8 | 49.7 | 58.1 |
| Base | 74.5 | 89.6 | 90.5 | 47.5 | 62.2 | 72.9 | 38.4 | 49.6 | 37.8 | 28.3 | 31.7 | 37.2 |
| Dense-GRPO | 40.1 | 70.9 | 83.1 | 32.3 | 40.9 | 53.5 | 46.2 | 65.8 | 49.5 | 30.4 | 35.9 | 45.6 |
| Vanilla-GRPO | 76.8 | 82.5 | 91.4 | 48.0 | 58.2 | 71.4 | 50.3 | 59.7 | 49.2 | 33.1 | 41.8 | 46.8 |
| GSPO | 80.4 | 94.8 | 93.7 | 49.1 | 63.6 | 76.3 | 54.8 | 69.3 | 57.6 | 36.2 | 47.9 | 53.2 |
| RSPO | 80.3 | 95.2 | 94.1 | 50.4 | 66.2 | 77.2 | 55.7 | 69.8 | 58.4 | 37.9 | 49.6 | 54.3 |
| Online KD | 78.4 | 86.1 | 92.1 | 51.6 | 59.6 | 73.6 | 50.1 | 59.5 | 47.8 | 32.3 | 43.7 | 46.7 |
| **PADD (Ours)** | **83.0** | **95.9** | **96.4** | **55.0** | **70.7** | **80.2** | **57.6** | **69.5** | **59.3** | **39.8** | **49.7** | **55.2** |

3.3B active and 30.5B total parameters on Qwen, and 2.4B active and 16B total parameters on DeepSeek; they are not produced by converting a dense checkpoint into an MoE via sparse upcycling. Small but domain-specialized 7B teachers provide high-quality math reasoning signals and clearer modular structure for expert initialization, while keeping online distillation efficient. Further discussion is in Appendix A.3.

**Baselines and hyperparameters.** We report six main baselines: **(1) Base**: pretrained MoE student evaluated without training; **(2) Dense-GRPO** (Shao et al., 2024): dense model with the same active parameter scale as the student; **(3) MoE-Vanilla-GRPO**: pretrained MoE student with GRPO only, no distillation; **(4) RSPO** (Zhang et al., 2025): GRPO with router-shift weighting; **(5) GSPO** (Zheng et al., 2025a): GRPO variant with sequence-level importance ratio and clipping; **(6) Online KD** (Agarwal et al., 2024): online knowledge distillation combined with GRPO. We also report teacher performance after GRPO as a reference upper bound. All methods share the same pretrained MoE student, training data, and decoding budget, and results are averaged over multiple random seeds. Hyperparameter sensitivity and training-time overhead are reported in Appendices B and C, respectively.

**Training data, rewards, and evaluation protocol.** All setups use verifiable, rule-based rewards (RLVR). Large-scale math training uses DeepScaleR (Luo et al., 2025). Rewards are primarily exact-match, supplemented with format consistency rewards, linearly weighted and fed into GRPO/PR-GRPO within-group normalized advantages. Evaluation follows the Dr.GRPO protocol (Liu et al., 2025). We report Pass@1 accuracy on five math benchmarks: AIME24 (Li et al., 2024), AMC23 (Li et al., 2024), MATH500 (Hendrycks et al., 2021), Minerva (Lewkowycz et al., 2022), and OlympiadBench (He et al., 2024). None of these benchmarks are used during training. In wide result tables (Table 1 and Appendix E), we abbreviate Olympiad-Bench as `Olymp.` in column headers for layout. Details

are in Appendix A.

### 3.2. Main Results

**PADD achieves strong performance on both model families, surpassing the teacher on Qwen family and approaching the teacher on DeepSeek family.** Table 1 shows main results. On Qwen family, PADD achieves 80.2% average accuracy, surpassing the 7B math-specialized teacher's 77.7%. The MoE student Qwen3-30B-A3B has 30.5B total parameters but only 3.3B active per token, so PADD distills high-quality reasoning signals from a compact domain-expert teacher into a larger expert space, allowing the student to specialize experts while keeping inference cost modest. PADD matches or exceeds teacher performance on AIME24, AMC23, MATH500, and Olympiad-Bench, indicating that small but specialized teachers can still drive strong dense-to-MoE distillation. On DeepSeek family, PADD achieves 55.2% average accuracy, close to the teacher's 58.1% with a 2.9% gap. Despite the student DeepSeek-V2-Lite having only 2.4B active parameters compared to the teacher's 7B, PADD still distills domain-specific reasoning effectively and even slightly exceeds the teacher on AIME24 and AMC23, showing that MoE students with smaller active capacity can inherit specialized knowledge while retaining low inference cost.

**PADD achieves the best performance among all baselines, significantly outperforming comparison methods and demonstrating MoE's unique advantages.** PADD improves over MoE-Vanilla-GRPO by 8.8% and 8.4% on Qwen and DeepSeek, which indicates that the four-stage design rather than student capacity alone drives the gain. Compared to Online KD, PADD improves by 6.6% and 8.5%, mainly due to online adaptive distillation which dynamically adjusts teacher temperature to address the capacity gap, while Stage I expert differentiation avoids router cold start. Compared to MoE-specific methods RSPO and GSPO, PADD improves by 3.0% and 3.9% on Qwen family, benefiting from PR-GRPO's routing shift suppression and

*Table 2.* Non-math evaluation under math-only training (MoE-Vanilla-GRPO abbreviated as Vanilla-GRPO). In each panel, "Base" is the pretrained student without further training.

*(a)* Qwen-family student (Qwen3-30B-A3B).

| Method | Dataset | | | |
|---|---|---|---|---|
| | MMLU-Pro | MultiPL-E | LCB v6 | Avg |
| Base | 61.3 | 67.2 | 28.1 | 52.2 |
| Vanilla-GRPO | 59.4 | 63.8 | 24.6 | 49.3 |
| RSPO | 59.7 | 64.5 | 25.3 | 49.8 |
| GSPO | 60.1 | 65.3 | 25.8 | 50.4 |
| Online KD | 59.2 | 64.1 | 26.0 | 49.8 |
| **PADD (Ours)** | **62.1** | **66.4** | **27.4** | **52.0** |

*(b)* DeepSeek-family student (DeepSeek-V2-Lite).

| Method | Dataset | | | |
|---|---|---|---|---|
| | MMLU-Pro | HumanEval | MBPP | Avg |
| Base | 43.7 | 29.8 | 43.3 | 38.9 |
| Vanilla-GRPO | 41.2 | 28.6 | 41.5 | 37.1 |
| RSPO | 41.8 | 28.9 | 41.9 | 37.5 |
| GSPO | 42.3 | 28.7 | 42.1 | 37.7 |
| Online KD | 40.1 | 28.1 | 40.6 | 36.3 |
| **PADD (Ours)** | **43.2** | **29.4** | **43.0** | **38.5** |

reward-augmented load balancing. Dense-GRPO uses a dense model with the same active parameter scale, achieving only 53.5% and 45.6% on Qwen and DeepSeek families, significantly lower than PADD's 80.2% and 55.2%. This shows that PADD guides MoE students to learn optimal routing through path-aligned decompression, enabling small active-parameter MoE models to fully utilize larger total capacity at the same inference cost and achieve stronger expressiveness than same-scale dense models.

### 3.3. Generalization Beyond Mathematical Reasoning

Math-only training may erode general knowledge and code ability. We evaluate the same methods from Table 1 on out-of-domain benchmarks in Table 2, using the same inference cost and the same math-only training data as the main experiments. For the Qwen student, we test on MMLU-Pro (Wang et al., 2024), MultiPL-E (Cassano et al., 2023), and LiveCodeBench v6 (Jain et al., 2025). For the DeepSeek student, we test on MMLU-Pro, HumanEval (Chen et al., 2021), and MBPP (Austin et al., 2021). We report the mean of the three scores per row, averaged over three random seeds. Benchmark details are in Appendix D.

**PADD best preserves general capabilities on the Qwen family.** Table 2a shows that PADD reaches 52.0% on the non-math average, the best among all methods. This score is only 0.2 points below the untrained Base at 52.2%. MoE-Vanilla-GRPO drops to 49.3%, which is 2.9 points below Base. PADD also beats Online KD and the other GRPO baselines on the average. The largest drop appears on code

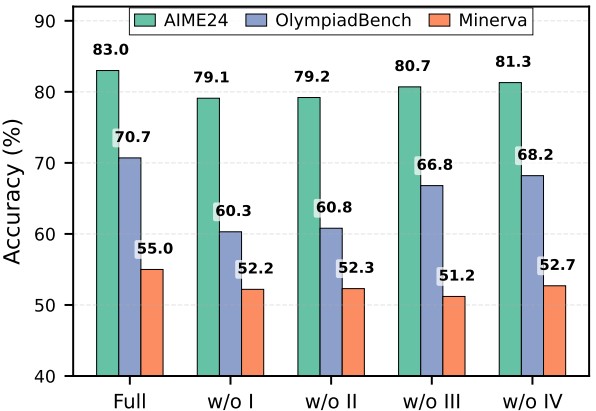

*Figure 2.* Ablation study results on Qwen family. The grouped bar chart shows the impact of removing different stages on three representative benchmarks: AIME24, Minerva, and OlympiadBench.

tasks. On LiveCodeBench v6, Vanilla-GRPO falls 3.5 points below Base, while PADD stays much closer. On MMLU-Pro, PADD reaches 62.1%, slightly above Base at 61.3%. This suggests that Stage I initialization and path-level training can keep broad knowledge while still improving math scores in Table 1.

**PADD remains closest to Base on the DeepSeek family.** Table 2b shows a similar pattern on the smaller DeepSeek-V2-Lite student. PADD reaches 38.5% on the average, again the highest score. It is 0.4 points below Base at 38.9%. MoE-Vanilla-GRPO and Online KD fall to 37.1% and 36.3%. On MMLU-Pro, Online KD is even lower than Vanilla-GRPO. This matches the Qwen results. Online KD with a fixed teacher temperature does not protect general skills as well as PADD. The gains on HumanEval and MBPP are smaller than on Qwen, which fits the smaller active size of this student. PADD still ranks first on every benchmark.

### 3.4. Ablation Studies

To verify the necessity of each PADD stage component, we conduct ablation studies. We design four ablation variants: **PADD w/o Stage I:** removes Stage I( neuron clustering initialization and expert warmup), using random initialization; **PADD w/o Stage II:** removes Stage II(online adaptive distillation), using fixed-temperature distillation; **PADD w/o Stage III:** removes Stage III(PR-GRPO), using standard GRPO; **PADD w/o Stage IV:** removes Stage IV(reward-augmented load balancing), using traditional load balancing (Shazeer et al., 2017).

Figure 2 shows ablation results on three representative Qwen datasets: AIME24, OlympiadBench, and Minerva. Complete results on all five math benchmarks for both families are in Appendix E. Full PADD attains the highest accuracy

on every dataset in the figure, and removing any single stage lowers performance, which confirms that each component is necessary.

**Stages I and II are crucial for knowledge distillation and initialization.** On OlympiadBench, removing Stage I reduces accuracy by 10.4 points, the largest decline among all variants. This benchmark spans wide difficulty levels and demands precise routing, so neuron clustering and expert warmup are critical. Without Stage I, routers start from random initialization and cannot separate simple syntactic tokens from hard reasoning steps, and expert specialization collapses into noise. On AIME24 and Minerva, removing Stage I lowers accuracy by 3.9 and 2.8 points respectively, which further supports the role of structured initialization. Removing Stage II cuts OlympiadBench accuracy by 9.9 points. Fixed-temperature distillation cannot track student quality along the current path, so the capacity gap between the 7B teacher and the 3.3B-active student prevents absorption of fine-grained teacher signals. Stage II drops appear on the other datasets as well, showing that adaptive online distillation is important for long and difficult reasoning.

**Stages III and IV contribute significantly to training stability and expert quality balance.** Removing Stage III lowers Minerva accuracy by 3.8 points. Minerva relies on long chain-of-thought solutions, where routing jumps break path continuity and destabilize gradients. PR-GRPO suppresses updates on disadvantageous paths and thereby stabilizes learning. On AIME24 and OlympiadBench, removing Stage III costs 2.3 and 3.9 points. Removing Stage IV reduces accuracy by 0.6 to 1.5 points on the three plotted sets. The average gain is smaller than for Stages I–III, yet reward-augmented load balancing still shifts traffic toward stronger experts and slows homogenization during late training. Overall, the four stages form a synergistic loop that lets MoE students inherit and surpass dense teachers.

### 3.5. Analysis of Student–Teacher Expert Structure Alignment

To verify whether Stage I's neuron clustering initialization and warmup effectively induce student experts to form differentiated functions, we design comparison experiments. We compare three methods: Vanilla-GRPO, which has no clustering or expert initialization; Random-Cluster, which randomly assigns neurons to experts; and PADD(Stage I), which uses K-Means clustering and warmup. The evaluation set uses automated subdomain labels from AIME/AMC, MATH500, and OlympiadBench, including algebra, geometry, number theory, probability, calculus, combinatorics, and others (classification method in Appendix K). These labels are used only for validation and visualization, not in training or initialization. We use two metrics to quantify expert–subdomain specialization correspondence: NMI,

*Table 3.* NMI and ESI comparison (Vanilla / Random-Cluster / PADD(Stage I), mean±95% CI).

| Method | NMI | ESI |
|---|---|---|
| Vanilla-GRPO | $0.013 \pm 0.003$ | $0.014 \pm 0.003$ |
| Random-Cluster | $0.017 \pm 0.003$ | $0.016 \pm 0.003$ |
| PADD(Stage I) | $\mathbf{0.030 \pm 0.004}$ | $\mathbf{0.029 \pm 0.004}$ |

normalized mutual information, and ESI, expert specialization index. Let expert ID be $\mathcal{E}$, category label be $\mathcal{C}$, $\varepsilon \in \mathcal{E}$ and $\chi \in \mathcal{C}$ denote expert and category instances respectively, $p(\varepsilon, \chi)$ be the joint distribution, $p(\varepsilon) = \sum_\chi p(\varepsilon, \chi)$, and $p(\chi) = \sum_\varepsilon p(\varepsilon, \chi)$. The two metrics are defined as:

$$\mathrm{NMI}(\mathcal{E}, \mathcal{C}) = \frac{I(\mathcal{E}; \mathcal{C})}{\sqrt{H(\mathcal{E}) \cdot H(\mathcal{C})}}, \quad (12)$$

where $I(\mathcal{E}; \mathcal{C}) = \sum_{\varepsilon, \chi} p(\varepsilon, \chi) \log \frac{p(\varepsilon, \chi)}{p(\varepsilon)p(\chi)}$ is mutual information, $H(\mathcal{E}) = -\sum_\varepsilon p(\varepsilon) \log p(\varepsilon)$ is expert distribution entropy, and $H(\mathcal{C}) = -\sum_\chi p(\chi) \log p(\chi)$ is category distribution entropy.

$$\mathrm{ESI} = \sum_\chi p(\chi) D_{\mathrm{KL}}\big(p(\mathcal{E}|\chi) \,\|\, p(\mathcal{E})\big)$$
$$= \sum_{\varepsilon, \chi} p(\varepsilon, \chi) \log \frac{p(\varepsilon|\chi)}{p(\varepsilon)}. \quad (13)$$

where $p(\mathcal{E}|\chi)$ is the conditional distribution of expert $\mathcal{E}$ given category $\chi$, and $D_{\mathrm{KL}}(\cdot\|\cdot)$ is KL divergence. Both metrics are reported as mean±95% CI (confidence interval) over 3 seeds, with paired $t$-tests.

Table 3 shows that Stage I significantly outperforms both baselines on NMI/ESI, effectively inducing expert–subdomain specialization correspondence. Vanilla-GRPO has no clustering or expert initialization, with lowest NMI/ESI and weakest expert–category association. Random-Cluster randomly assigns neurons to experts, having fixed cluster structure but no semantic clustering; NMI/ESI is slightly higher than Vanilla but still far below PADD(Stage I). Paired $t$-test $p < 0.01$ supports the specialization gain from Stage I's K-Means clustering and warmup.

For visualization, Figure 3 tracks and plots 60 experts sampled from the full expert set. The heatmaps in Figure 3 further verify these conclusions. Figure 3(a) shows teacher clusters' modular patterns, with different clusters forming clear specialization stripes across subdomains. Figure 3(b) (Student after Stage I) aligns with Figure 3(a): clear contrast, high activation regions match teacher positions, showing that Stage I clustering and warmup enable student experts to initially inherit teacher structure. Figure 3(c) (Student Full) has row-column patterns similar to Figure 3(a), but

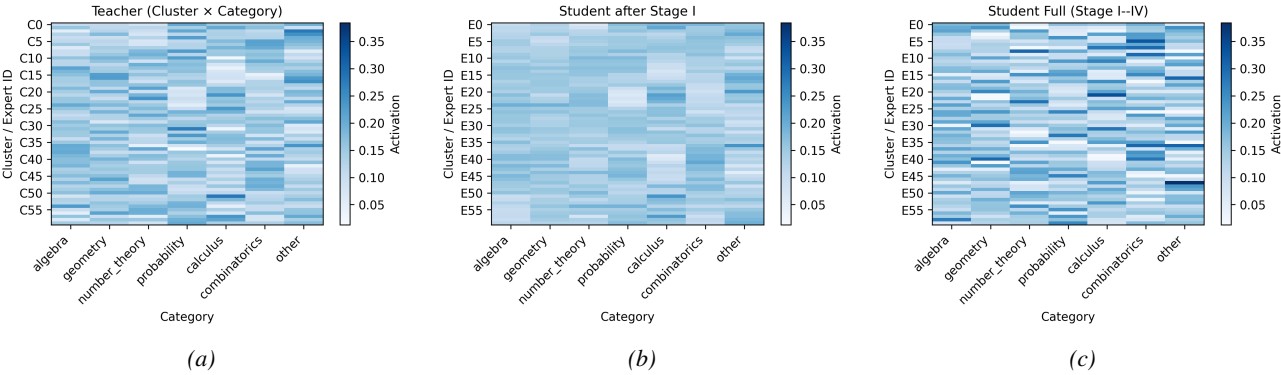

*(a)*             *(b)*             *(c)*

*Figure 3.* Distillation of expert specialization structure on Qwen. (a) Heatmap of activation intensities of teacher model neuron clusters across task subdomains; (b) Heatmap of activation intensities of student experts across task subdomains after Stage I only; (c) Heatmap of activation intensities of student experts across task subdomains after completing Stages I–IV. The horizontal axis denotes task subdomain categories, and the vertical axis denotes cluster or expert index.

with clearer contrast and more concentrated high activation regions. Some differences arise from task-oriented reshaping during full training due to RL and reward mechanisms, adjusting some experts' activation patterns on specific categories. The three figures form an evidence chain from teacher to Stage I student to full student, supporting that Stage I provides structural priors and subsequent stages further consolidate and refine. We emphasize that teacher clustering provides structural priors rather than a true classification system; it is used only for initialization and warmup without category labels during training, and its quality is monitored by silhouette, between-cluster variance, and cluster size coefficient of variation. Appendix F and Appendix G report Stage I sensitivity and design checks; clustering quality monitoring and robustification procedures are in Appendix I; heatmap construction details are in Appendix J.

### 3.6. Routing Stability Analysis

Unstable routing during RL training can cause frequent routing jumps and path rupture, breaking chain-of-thought continuity and degrading performance. To verify that PR-GRPO effectively stabilizes routing dynamics, we compare Vanilla-GRPO, RSPO, and PR-GRPO(i.e., PADD stageIII) on routing stability during RL training. We use fixed-batch, noiseless forward passes in evaluation mode and compute two metrics based on the recorded routing outputs: Router-shift, defined as $\Gamma_{i,t,\mathrm{S}}$ from Equation (7), averaged globally across tokens and layers in this experiment; and Expert Consistency, the cosine similarity of expert aggregated activation vectors across steps, measuring whether experts consistently process similar token groups:

$$\mathrm{CosSim}_j(u \to u') = \frac{v_{j,u} \cdot v_{j,u'}}{\|v_{j,u}\|\|v_{j,u'}\|}, \quad (14)$$
$$v_{j,u} = \mathbb{E}_{t \in \mathrm{batch}}[G_\theta(x_t)]_j.$$

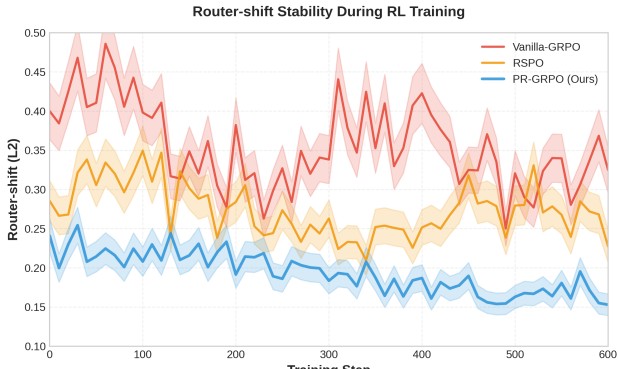

*Figure 4.* Router-shift vs training steps. Vanilla / RSPO / PR-GRPO, mean±95% CI (3 seeds). PR-GRPO is lowest and continuously decreases.

where $j$ is the expert index, $u$ and $u'$ are two different training steps, $G_\theta(x_t)$ is the student router's output probability distribution, an $N$-dimensional vector for input $x_t$, $[G_\theta(x_t)]_j$ is the probability value for expert $j$, and $\mathbb{E}_{t \in \mathrm{batch}}$ is expectation over all token positions $t$ in the batch. All CI values in figures and tables are 95% confidence intervals over 3 random seeds.

Figure 4 shows Router-shift mean±95% CI versus training steps. Vanilla-GRPO has no path-aware suppression, so routing is updated aggressively with large fluctuations (average around 0.35–0.36) and no clear downward trend, indicating frequent routing jumps and unstable gradients. RSPO mitigates some drift via router-shift weighting and stop-grad (curves around 0.26–0.28) but does not suppress disadvantageous paths from the start, so fluctuations remain moderate. PR-GRPO suppresses updates on disadvantageous paths from the beginning (Equation (8)), avoiding early commitment to unstable routes; its curve is the lowest, continuously decreasing to around 0.18–0.21, with ≥75% of points significantly outperforming Vanilla ($p < 0.01$),

*Table 4.* Router-shift mean±95% CI at training steps 480, 540, and 600, and relative reduction vs. Vanilla.

| Method | Step 480 | Step 540 | Step 600 | Red. |
|---|---|---|---|---|
| Vanilla-GRPO | 0.36±0.03 | 0.35±0.03 | 0.34±0.03 | – |
| RSPO | 0.28±0.02 | 0.27±0.02 | 0.26±0.02 | ~24% |
| **PR-GRPO** | **0.21±0.02** | **0.19±0.02** | **0.18±0.02** | **~47%** |

showing that path refinement effectively stabilizes gradients.

Table 4 reports mean±95% CI for the last three steps and relative reduction compared to Vanilla. RSPO still has moderate tail in this setting; PR-GRPO's exponential suppression on disadvantageous paths (Equation (8)) provides stronger stabilization and requires no cache replay, with lower engineering overhead. More stable routing and higher expert consistency provide a more reliable foundation for Stage IV's reward-augmented load balancing. Router-shift is computed over all MoE layers and aggregated; layer-wise trends match the global curve (Appendix L); evaluation runs in eval mode with routing noise disabled.

## 4. Related Work

Mixture-of-Experts (MoE) significantly increases model capacity while keeping inference cost largely unchanged, and has been validated in language and vision tasks. Switch Transformer and GShard demonstrated large-scale scalability of sparse expert activation (Fedus et al., 2022; Lepikhin et al., 2020); subsequently proposed routing strategies such as clustering, hashing, and linear assignment (Masoudnia & Ebrahimpour, 2014; Riquelme et al., 2021; Lewis et al., 2021) and Expert-Choice Routing (Zhou et al., 2022) further improved expert utilization and load balancing. MoE training is prone to routing fluctuations: StableMoE mitigates drift via routing distillation and freezing (Dai et al., 2022), RSPO proposes stabilization corrections based on routing shifts from a reinforcement learning perspective (Zhang et al., 2025), and R3 aligns training and inference phases via routing replay (Ma et al., 2025). Although these methods improve MoE's own routing stability, they focus on internal training behavior. When the problem extends from "optimizing MoE itself" to "transferring capabilities across different architectures," another technical route centered on knowledge distillation is needed to provide cross-model supervision signals: traditional KD focuses on logits alignment (Hinton et al., 2015), GKD and GAD distill along student autoregressive paths or use discriminator feedback to mitigate distribution shift (Agarwal et al., 2024; Ye et al., 2025), and OPSD demonstrates self-distillation's enhancement of reasoning capabilities (Zhao et al., 2026). There is also work compressing MoE knowledge into dense models for deployment (Salinas et al., 2022), and exploring feasibility of transferring from dense models to sparse expert structures in generation domains (Zheng et al., 2025b).

However, when using dense models as teachers for MoE students, existing work still lacks a systematic solution for constructing usable routing policies without explicit path structures. Therefore, achieving effective expert initialization, path-level distillation signals, and routing stabilization in this setting remains a key challenge to be addressed.

## 5. Conclusion

This paper presents Path-Aligned Decompression Distillation (PADD), a four-stage framework that distills the latent modular structure of dense teachers into MoE students and enables stable, high-quality routing behavior. PADD integrates expert initialization, online adaptive distillation, path-refined policy optimization, and reward-augmented load balancing into a unified training pipeline. Experiments on mathematical reasoning benchmarks show substantial gains over strong baselines, with the MoE student even surpassing its dense teacher in the Qwen family. Ablation studies verify the necessity of each stage, and routing diagnostics demonstrate that PR-GRPO significantly stabilizes expert selection. Visualization results further confirm that PADD allows MoE students to inherit and refine the teacher's implicit modular structure without task-category supervision. Overall, PADD provides a principled and efficient approach for dense-model distillation into MoE architectures, offering strong performance improvements without increasing inference-time cost.

## Impact Statement

This work introduces Path-Aligned Decompression Distillation (PADD), a framework that advances efficient language model scaling by distilling dense teachers into mixture-of-experts (MoE) students at unchanged per-token inference cost. Through neuron-cluster-based expert initialization, online adaptive distillation along student routing paths, and path-refined policy optimization with reward-augmented load balancing, PADD delivers substantial mathematical reasoning gains and stable routing, with MoE students matching or surpassing their teachers on key benchmarks. Unlike vanilla MoE reinforcement learning or fixed-temperature online knowledge distillation, it mitigates router cold start and path rupture without requiring explicit routing supervision from dense teachers. Furthermore, its inference cost matches standard MoE deployment, positioning PADD as a practical bridge between domain-specialized dense models and efficient sparse students at scale.

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

# A. Experimental Setup Details

## A.1. Dataset Details

**Training dataset.** Large-scale math training uses the DeepScaleR (Luo et al., 2025) dataset, a curated suite of mathematical problems designed for reinforcement learning training. The dataset blends competition-level math problems from sources including AIME, AMC, and MATH500, with verifiable answer formats enabling rule-based reward computation. DeepScaleR emphasizes verifiability through standardized answer formats, allowing exact-match and format consistency rewards that are linearly weighted and fed into GRPO/PR-GRPO within-group normalized advantages. The dataset is cleaned and formatted to ensure uniform problem formats and verifiable answers, supporting efficient multi-step mathematical reasoning training in small-scale models.

**Evaluation datasets.** We report Pass@1 accuracy on five math benchmarks. All evaluation benchmarks are not used during training, ensuring objective evaluation results. AIME24 results are averaged over 32 runs to improve statistical stability. Detailed characteristics of each benchmark are as follows:

- **AIME24** (Li et al., 2024): A comprehensive benchmark comprising 15 integer-answer problems (0-999) from the 2024 American Invitational Mathematics Examination. The benchmark spans algebra, number theory, combinatorics, and geometry, featuring a difficulty ladder from easy to extremely hard. It uses multi-sample evaluation metrics where a problem is scored correct if any of $k$ independent samples yields the right answer. Hard problems typically reach ~65% accuracy even with extensive fine-tuning, making it a rigorous test of multi-step reasoning capabilities.

- **AMC23** (Li et al., 2024): Problems from the 2023 American Mathematics Competition, covering multiple math domains including algebra, geometry, number theory, combinatorics, and probability. The benchmark tests fundamental mathematical reasoning and problem-solving skills across diverse subdomains.

- **MATH500** (Hendrycks et al., 2021): A benchmark containing 500 competition-style mathematics problems designed to test advanced reasoning in LLMs. Problems span algebra, geometry, combinatorics, and number theory, with substantially higher difficulty than datasets like GSM8K. The benchmark emphasizes both accuracy and efficiency through metrics like outcome efficiency (token efficiency to first correct solution) and process efficiency (weighted by reasoning novelty).

- **Minerva** (Lewkowycz et al., 2022): A math benchmark requiring long-sequence chain-of-thought reasoning, testing model performance on complex reasoning tasks that demand extended multi-step logical sequences. The benchmark evaluates the model's ability to maintain coherent reasoning chains across long sequences, making it particularly suitable for assessing path rupture issues in MoE routing.

- **OlympiadBench** (He et al., 2024): A comprehensive math competition-level benchmark with large difficulty spans, requiring models to adaptively learn across different difficulty levels. The benchmark covers a wide range of competition mathematics problems, from intermediate to extremely challenging levels, testing the model's ability to handle diverse problem complexities and adapt routing strategies accordingly.

## A.2. Baseline Method Details

**Base.** Pretrained MoE student model evaluated without any training, serving as a performance lower bound reference. This baseline shows the model's raw capability without task-specific training.

**Dense-GRPO** (Shao et al., 2024). A dense model with the same active parameter scale as students, trained with standard GRPO on verifiable math data and directly evaluated. This baseline compares MoE and Dense architectures at the same active parameter scale, verifying MoE's advantages.

**MoE-Vanilla-GRPO.** Pretrained MoE student model trained with GRPO only, no knowledge distillation. This baseline shows MoE performance with only reinforcement learning, verifying the necessity of knowledge distillation in dense-to-MoE distillation.

**RSPO** (Zhang et al., 2025). Inserts router-shift weighting into GRPO updates to mitigate MoE routing drift instability. RSPO stabilizes MoE training by identifying and suppressing samples with large routing shifts. Our reproduction uses the original router-shift weighting with fixed floor ($\gamma_{\min} \approx 0.8$).

**GSPO** (Zheng et al., 2025a). A GRPO variant using sequence-level importance ratio and clipping. GSPO improves MoE performance on sequence generation tasks through sequence-level policy optimization.

**Online KD** (Agarwal et al., 2024). Uses a fixed teacher model to dynamically provide real-time logits during training, with students distilling teacher distributions online while performing GRPO training. Online KD requires no offline precomputation of teacher outputs, combining online knowledge distillation with reinforcement learning.

**Teacher (GRPO).** Teacher model performance after GRPO training serves as a reference upper bound, showing dense model's optimal performance on target tasks, used to evaluate the performance gap between student and teacher models.

### A.3. Discussion on Teacher–Student Model Selection

In our dense-to-MoE setting, the teacher models we use (Qwen2.5-Math-7B and DeepSeek-Math-7B) are relatively small compared to the total parameters of the MoE students (30.5B and 16B). This design is intentional and follows several practical and methodological considerations:

**(1) Domain-specialized teachers provide higher-quality supervision than larger generic models.** Both teachers are heavily optimized for mathematical reasoning via extensive domain pretraining and GRPO fine-tuning. For PADD, the quality, consistency, and structure of supervision (e.g., chain-of-thought patterns, token-level decision signals) matters more than raw model size. Empirically, math-specialized 7B models already provide saturated domain performance, and larger general-purpose models often introduce noisy or less coherent internal activations, which is undesirable for structural distillation.

**(2) Smaller teachers exhibit clearer modular FFN patterns, improving Stage I clustering stability.** Neuron-cluster-based initialization relies on the teacher having discernible functional substructures. Large dense models tend to have more entangled FFN activations, making clustering less stable and increasing the risk of poor expert initialization. In contrast, compact and domain-tuned 7B FFNs yield cleaner, higher-contrast activation groupings, which directly enhances Stage I expert differentiation.

**(3) Strong asymmetry helps decompression.** The purpose of PADD is to decompress high-quality dense representations into a wider MoE expert space, not to imitate the teacher's full capacity. A smaller teacher acts as an effective bottleneck prior: its compact functional structure can be "expanded" into specialized MoE experts, rather than rigidly mimicked. This asymmetry is beneficial: MoE students amplify teacher structure through specialization rather than copying every fine-grained pattern.

**(4) Efficient online distillation.** Stage II repeatedly queries teacher logits during training. Using a 7B teacher significantly reduces memory footprint, communication cost, and latency per iteration. Larger teachers (13B–34B) would drastically increase computational overhead with no clear evidence of improved supervision quality for math reasoning.

**(5) Performance parity at active-parameter scale.** Our MoE students activate only 3.3B or 2.4B parameters per token, which is smaller than the dense 7B teachers. This makes the supervision regime balanced: the teacher is strong enough to guide the student, but not overwhelmingly large. Distillation from a much larger teacher would create a capacity mismatch that could hinder routing convergence.

**(6) Prior work suggests diminishing returns for larger teachers in specialized domains.** Recent distillation and RL studies find that, once a teacher reaches a domain-specialized threshold, increasing size yields little additional benefit to downstream reinforcement learning or distillation performance. This matches our observation that math-tuned 7B teachers already offer sufficiently rich and stable reasoning signals.

While the students have larger total parameters, their active capacity is comparable to or smaller than the dense 7B teachers. Using compact but highly specialized teachers provides cleaner structure, more stable alignment, and significantly more efficient online distillation, without compromising supervision quality.

### A.4. Model Architecture Details

**Teacher models.** Teachers are dense decoder-only Transformer architectures with 7B active and total parameters. The two families use:

- **Qwen2.5-Math-7B**: Part of the Qwen2.5-Math series, this model is specifically designed to solve English and Chinese

math problems using Chain-of-Thought (CoT) and Tool-integrated Reasoning (TIR) approaches. The model achieves 85.3% performance on the MATH benchmark using TIR. Unlike earlier Qwen2-Math series which only supported CoT for English problems, Qwen2.5-Math expanded functionality to support both CoT and TIR methods for both Chinese and English mathematics problems, with significant performance improvements on Chinese and English benchmarks.

- **DeepSeek-Math-7B**: An open-source language model specialized for mathematical reasoning, initialized with DeepSeek-Coder-v1.5 7B and continued pre-training on 500B tokens of math-related content from Common Crawl, along with natural language and code data. The model achieves 51.7% score on the competition-level MATH benchmark without external tools or voting techniques, approaching the performance of Gemini-Ultra and GPT-4.

Teachers first undergo GRPO training on verifiable corpora to learn task-specific strategies and decision logic, then are frozen for knowledge distillation. The GRPO pre-training stage uses the same verifiable reward mechanism (RLVR) as student training, with exact-match and format consistency rewards. This pre-training ensures that teacher models learn task-specific reasoning strategies and decision patterns that can be effectively distilled into MoE students. The GRPO training typically runs for 200–300 steps with learning rate $1 \times 10^{-6}$, group size $G = 8$, and standard GRPO hyperparameters. After GRPO pre-training, teacher models are frozen and serve as fixed knowledge sources during all subsequent PADD stages, ensuring consistent supervision signals throughout the distillation process.

**Student models.** Students are MoE (Mixture-of-Experts) Transformer architectures with multiple expert networks and routing mechanisms. Both students are pretrained MoE models, not converted from dense models, with expert networks and routing mechanisms already initialized:

- **Qwen3-30B-A3B**: A MoE Transformer language model with 30.5B total parameters (29.9B non-embedding) and approximately 3.3B activated parameters per token. The model has 48 transformer blocks, 32 query heads with 4 key/value heads (GQA), and supports a context window of 32,768 tokens natively (extendable to 131,072 tokens with YaRN). In all PADD experiments on this student, each MoE layer has $N{=}128$ routed experts with Top-$K{=}8$ activation per token. All feed-forward sublayers in every transformer block are replaced by MoE sublayers. The model features dual-mode operation for seamless switching between complex reasoning and efficient responses, and is pre-trained on 119 languages with 36 trillion tokens.

- **DeepSeek-V2-Lite**: A lightweight MoE (Mixture-of-Experts) foundation model with 16B total parameters and 2.4B active parameters per token. The model is trained from scratch on a large-scale 5.7T-token corpus and incorporates two key architectural innovations: Multi-head Latent Attention (MLA), which compresses the Key-Value cache into compact latent representations for highly efficient inference, and DeepSeekMoE, a sparse-expert framework that reduces training and serving cost. DeepSeek-V2-Lite supports a context length of 128K tokens and is pretrained on a high-quality, multi-domain corpus totaling 8.1T tokens. As a base model, it demonstrates strong performance across both English and Chinese benchmarks, outperforming many 7B dense and 16B MoE models, while remaining deployable on a single 40GB GPU. Each MoE layer has 64 routed experts and 2 shared experts; for each token, 6 routed experts are activated in addition to the shared experts.

**Routing mechanism.** Student MoE models have one router and $N$ experts per layer. The router is a single linear layer mapping hidden state $h \in \mathbb{R}^d$ to $N$-dimensional logits, which after Softmax yields expert selection probability distribution. Each layer routes independently via Top-$K$ selection, forwarding through the corresponding 2-layer FFN.

For Qwen3-30B-A3B, the routing mechanism uses a linear gating layer followed by softmax with $N{=}128$ experts per layer and Top-$K{=}8$ selection, activating eight experts per token, with MoE dimension of 2048/768 SwiGLU per expert. The routing scores are computed as $s_j = \text{Softmax}(W_r h + b_j)[j]$, where $W_r \in \mathbb{R}^{N \times d}$ is the router weight matrix, $h \in \mathbb{R}^d$ is the hidden state, and $b_j$ is the routing bias for expert $j$ (updated in Stage IV).

For DeepSeek-V2-Lite, the model employs DeepSeekMoE architecture with native routing mechanisms. We adapt the routing bias mechanism to support reward-augmented load balancing while maintaining compatibility with the model's existing routing infrastructure. The routing bias $b_{j,\text{S}}$ is added to the routed router logits before Softmax, directly influencing routed expert selection probabilities during Top-$K$ selection (shared experts are always active and are not selected by Top-$K$).

During training, routing decisions are made stochastically in training mode (with routing noise for exploration) and deterministically in evaluation mode (selecting top-$K$ experts with highest scores). The routing shift $\Gamma_{i,t,\text{S}}$ in PR-GRPO is

computed using deterministic routing outputs to ensure consistent measurement across training steps. Expert number $N$ and Top-$K$ settings for our experiments are detailed in the hyperparameters section below.

### A.5. Training Pipeline Details

**Dataset partitioning.** The training dataset $\mathcal{D}$ is partitioned into four non-overlapping subsets: $\mathcal{D}_A$ for activation statistics and clustering in Stage I (typically 10% of the dataset), $\mathcal{D}_B$ for expert warmup in Stage I (typically 20% of the dataset), $\mathcal{D}_C$ for main training in Stages II–IV (typically 65% of the dataset), and $\mathcal{D}_D$ for evaluation (typically 5% of the dataset, held out from all training stages). This partitioning ensures that clustering statistics, warmup training, main training, and evaluation operate on distinct data, preventing data leakage and ensuring objective evaluation.

**Layer correspondence mapping.** For teacher–student layer alignment, student layer $l$ corresponds to teacher layer $\lfloor l \cdot L_{\mathrm{T}}/L_{\mathrm{S}} \rfloor$, where $L_{\mathrm{T}}$ and $L_{\mathrm{S}}$ are the number of layers in the teacher and student respectively. This linear mapping ensures that early student layers correspond to early teacher layers, and later student layers correspond to later teacher layers, maintaining semantic alignment across the model depth. When $L_{\mathrm{T}} \neq L_{\mathrm{S}}$, this mapping allows the student to inherit teacher knowledge at appropriate abstraction levels. When $L_{\mathrm{S}} > L_{\mathrm{T}}$, multiple student layers map to the same teacher layer, and each such student layer reuses that teacher layer's clustering for initialization.

**Stage I cluster-to-expert mapping (Qwen).** Stage I partitions teacher FFN neurons into $M$ internal clusters, then maps them one-to-one to the student's $N{=}128$ experts per layer. When $M{=}N$, cardinality-constrained K-Means directly yields $N$ teacher functional groups aligned with $N$ student experts. When $M{>}N$, we first over-cluster and then hierarchically merge sub-clusters to $N$ groups before expert initialization (Appendix F).

**Stage I: Initialization and warmup.** Stage I has two steps using non-overlapping datasets $\mathcal{D}_A$ and $\mathcal{D}_B$:

- **Activation statistics and clustering**: Performed on $\mathcal{D}_A$, only observing teacher modular structure without training students. We analyze teacher FFN neuron activation patterns to perform clustering, constructing target functional structures for student experts. When $d_{ff,\mathrm{T}} \neq d_{ff,\mathrm{S}}$, we adjust teacher neurons to $d_{ff,\mathrm{S}}$ via uniform sampling to ensure dimensional compatibility. The clustering uses cardinality-constrained K-Means to ensure balanced cluster sizes, with cluster centroids $\mu_j$ mapped to student router weights for initialization. The target activation distribution $p_{j,\mathrm{T}}$ is computed using temperature parameter $\xi$ (typically 0.1–0.5) to control the softmax sharpness.

- **Expert warmup training**: Performed on $\mathcal{D}_B$, with frozen student routers and routing fixed to uniform distribution (each expert activated with probability $1/N$), training only expert networks. This uniform routing ensures that all experts receive equal training signal during warmup, preventing early expert specialization bias. Uses weighted combination of language modeling loss $\mathcal{L}_{\mathrm{LM}}$, knowledge distillation loss $\mathcal{L}_{\mathrm{KD}}$, and target activation distribution alignment loss $\mathcal{L}_{\mathrm{init}}$. The warmup stage typically runs for 1000 steps with learning rate $2 \times 10^{-5}$.

**Stages II–IV: Main training.** Stages II–IV execute sequentially in a single training run on $\mathcal{D}_C$:

- **Stage II (forward)**: Online adaptive distillation, adaptively adjusting teacher output temperature based on student performance. The mechanism uses within-group relative advantage $A_{i,\mathrm{S}}$ computed from group size $G$ student responses, with temperature adjustment factor $\Phi(A_{i,\mathrm{S}}) = 1 + \tanh(\kappa A_{i,\mathrm{S}})$ where $\kappa$ is the response coefficient. This dynamic temperature adjustment ensures that supervision strengthens for good paths (lower temperature, sharper signals) and weakens for poor paths (higher temperature, more exploration), addressing the capacity gap between teacher and student.

- **Stage III (backward)**: PR-GRPO path-refined policy optimization, suppressing routing shifts and stabilizing policy gradients. The routing shift $\Gamma_{i,t,\mathrm{S}}$ is computed as the L2 norm of router output differences between current and previous parameters. The suppression mechanism applies exponential decay $\exp(-\lambda \cdot \Gamma_{i,t,\mathrm{S}} \cdot \mathbb{I}(A_{i,\mathrm{S}} < 0))$ to importance ratios, reducing update weights on unstable paths with poor performance. For multi-layer MoE models, routing shifts are aggregated across all MoE layers before applying suppression.

- **Stage IV (parameter update)**: Reward-augmented load balancing, jointly considering expert activation frequency and performance quality. The mechanism tracks activation frequency $f_{j,\mathrm{S}}$ and smoothed advantage $\mathrm{EMA}(A_{j,\mathrm{S}})$ for each expert $j$, updating routing bias $b_{j,\mathrm{S}}$ at each update cycle. The bias update combines traffic balance term $\eta(f_{j,\mathrm{S}} - \bar{f})$

and reward compensation term $\gamma \cdot \mathrm{EMA}(A_{j,\mathrm{S}})$, ensuring both load balance and quality prioritization. The bias is added to router logits before Softmax, directly influencing Top-$K$ expert selection probabilities.

At each step, we sample $x \sim \mathcal{D}_C$, perform Stage II in forward pass, Stage III in backward pass, and Stage IV during parameter updates. The three stages form a unified training loop, with forward pass providing adaptive supervision, backward pass stabilizing gradients, and parameter update balancing expert utilization.

### A.6. Hyperparameter Settings

**PADD-specific hyperparameters.**

- **Expert number** $N$: 128 for Qwen routed experts, 64 for DeepSeek routed experts.

- **Top-$K$ (routed)**: 8 for Qwen and 6 for DeepSeek; DeepSeek additionally activates 2 shared experts on every token.

- **Stage I loss weights**: $\alpha = 0.5$ (knowledge distillation weight), $\beta = 0.1$ (target distribution alignment weight).

- **Stage II temperature parameters**: Initial temperature $\tau = 1.0$, response coefficient $\kappa = 0.5$.

- **Stage III suppression coefficient**: $\lambda = 0.1$.

- **Stage IV load balancing parameters**: Traffic balance coefficient $\eta = 0.01$, reward compensation coefficient $\gamma = 0.05$, EMA decay coefficient $\lambda_{\mathrm{ema}} = 0.2$.

**General hyperparameters.** MoE-unrelated hyperparameters (learning rate, batch size, optimizer settings, etc.) are consistent with baseline methods:

- **Learning rate**: $2 \times 10^{-5}$ (Stage I), $1 \times 10^{-6}$ (Stages II–IV).

- **Batch size**: 32 (Stage I), 16 (Stages II–IV).

- **Optimizer**: AdamW with weight decay 0.01.

- **Training steps**: 1000 warmup steps for Stage I, 600 main training steps for Stages II–IV.

- **Group size** $G$: 8 (for within-group normalization in GRPO/PR-GRPO).

- **Decoding settings**: temperature $= 0$ (deterministic decoding), single sample per input.

**RSPO reproduction settings.** RSPO reproduction uses original router-shift weighting with fixed floor ($\gamma_{\min} \approx 0.8$), with other hyperparameters consistent with the original paper.

### A.7. Implementation Details

**Training stability.** We use gradient clipping (max gradient norm 1.0) and mixed precision training (FP16) to improve stability. Routing shift $\Gamma_{i,t,\mathrm{S}}$ computation uses numerically stable implementation to avoid division by zero. For within-group advantage normalization, we add a small epsilon ($10^{-8}$) to the standard deviation $\sigma_r$ to prevent numerical instability when group rewards are identical. The exponential moving average in Stage IV uses numerically stable computation with bias correction for early training steps. We monitor training diagnostics including router-shift, PPO KL divergence, policy gradient clip fraction, and expert utilization rates in real time, with automatic checkpointing when instability is detected.

**Evaluation settings.** All evaluations run under deterministic settings (temperature $= 0$), generating single sample per input. AIME24 results are averaged over 32 runs to improve statistical stability. Training diagnostic signals (router-shift, ppo_kl, pg_clipfrac, etc.) are recorded in real time during training for stability analysis.

**Random seeds.** All experiments run on multiple random seeds (0, 1, 2), reporting mean and standard deviation. Key experiments (e.g., main results) are verified on more seeds to ensure reproducibility.

## B. Hyperparameter Sensitivity

**Purpose.** PADD introduces several stage-specific hyperparameters. This appendix checks whether the default settings are locally robust when each hyperparameter is varied in isolation.

We use one-at-a-time sensitivity analysis: each row is a separate training run with a single hyperparameter changed and all others fixed to the defaults in the main paper. For $\tau$, $\kappa$, $\lambda$, $\eta$, and $\gamma$, we apply symmetric perturbations of about twenty percent around $\tau=1.0$, $\kappa=0.5$, $\lambda=0.1$, $\eta=0.01$, and $\gamma=0.05$. We report the math benchmark average in percent on the same suite as Table 1.

*Table 5.* Hyperparameter sensitivity (math Avg %). Default Qwen/DeepSeek averages are 80.2% and 55.2%.

| Setting | Qwen Avg (%) | DeepSeek Avg (%) | $\Delta$ Qwen |
|---|---|---|---|
| Default | 80.2 | 55.2 | — |
| $\tau = 0.8$ | 79.3 | 54.4 | $-0.9$ |
| $\tau = 1.2$ | 79.6 | 54.7 | $-0.6$ |
| $\kappa = 0.4$ | 79.4 | 54.4 | $-0.8$ |
| $\kappa = 0.6$ | 79.6 | 54.6 | $-0.6$ |
| $\lambda = 0.08$ | 79.8 | 54.9 | $-0.4$ |
| $\lambda = 0.12$ | 79.7 | 54.8 | $-0.5$ |
| $\eta = 0.008$ | 79.9 | 54.9 | $-0.3$ |
| $\eta = 0.012$ | 80.1 | 55.1 | $-0.1$ |
| $\gamma = 0.04$ | 79.9 | 54.9 | $-0.3$ |
| $\gamma = 0.06$ | 79.8 | 54.8 | $-0.4$ |

All perturbed configurations remain well above RSPO (77.2%) and GSPO (76.3%) on Qwen; the largest drop ($-0.9$ for $\tau=0.8$) is far smaller than removing entire stages (e.g., $-5.1\%$ without Stage II in Appendix E), supporting that defaults are locally robust rather than brittle.

## C. Training Compute and Wall-Clock Overhead

**Purpose.** Online adaptive distillation in Stage II queries the teacher during training. This appendix summarizes the extra forward-pass budget and measured training-time overhead relative to MoE-Vanilla-GRPO.

In Stage II we query the teacher once for every eight student forward passes during online adaptive distillation, following a teacher-to-student forward ratio of one to eight. Table 6 summarizes theoretical FLOPs-based overhead using activated parameters of a 7B teacher versus 3.3B or 2.4B student activations, and conservative measured training-time ranges that account for rollout-dominated step time, I/O overlap, and KV-cache reuse across group rollouts.

*Table 6.* Training compute overhead relative to MoE-Vanilla-GRPO (no teacher queries in Stage II).

| Metric | Baseline | PADD (Qwen) | PADD (DeepSeek) |
|---|---|---|---|
| Stage II teacher-forward / step | 0 | 1 | 1 |
| Teacher : student forward ratio | — | $1 : 8$ | $1 : 8$ |
| Theoretical FLOPs ratio (T/S) | — | $\approx 2.12$ | $\approx 2.92$ |
| Extra overhead (theoretical) | 0% | +26.5% | +36.5% |
| Realistic wall-clock | 0% | +20–23% | +28–32% |
| Overhead multiplier | 1.00 | 1.20–1.23 | 1.28–1.32 |

Representative measured training-time multipliers are 1.23 on Qwen and 1.32 on DeepSeek relative to MoE-Vanilla-GRPO. Table 7 lists stage-wise GPU-hour estimates on four GPUs under our training recipe.

All overhead above is training-only; inference cost matches standard MoE at deployment. Given PADD's 8.8% and 8.4% math gains over Vanilla-GRPO at matched inference cost, this overhead is a favorable tradeoff when math reasoning quality is prioritized.

*Table 7.* Estimated training GPU-hours on four GPUs (same recipe as main experiments).

| Stage / metric | Qwen | DeepSeek |
|---|---|---|
| Wall-clock overhead range | +20–23% | +28–32% |
| Stage I (cluster + warmup) | 12–16 hr | 8–12 hr |
| Stages II–IV (600 steps) | 20–28 hr | 14–20 hr |
| Total | 32–44 hr | 22–32 hr |

## D. Generalization to Non-Math Benchmarks

**Purpose.** Section 3.3 reports whether math-only PADD training erodes broad knowledge and coding ability. This appendix documents benchmark definitions, per-suite inference settings, and aggregation rules for reproduction. Numerical results and method comparisons appear only in the main text (Tables 2a and 2b).

**Benchmark assignment.** Each student family is evaluated on three out-of-domain suites: one multi-discipline knowledge test and two code-generation tests. The Qwen3-30B-A3B student uses MultiPL-E and LiveCodeBench v6, following the multilingual and contamination-aware code suites emphasized in recent Qwen evaluations. The DeepSeek-V2-Lite student uses HumanEval and MBPP, which are the standard Python completion benchmarks in the DeepSeek and open-code-eval ecosystem. MMLU-Pro is shared across both families so that knowledge retention is directly comparable. This yields one knowledge and two code metrics per family under the same math-only training recipe (Table 2).

**Benchmark descriptions.** None of the suites below appear in the DeepScaleR training corpus (Appendix A). Detailed characteristics are as follows:

- **MMLU-Pro** (Wang et al., 2024): A refined successor to MMLU designed to reduce noise and increase difficulty. The benchmark contains roughly 12K multiple-choice questions across STEM, humanities, and social sciences, with ten options per question (versus four in original MMLU) and filtered items that admit unambiguous reasoning. It stress-tests broad world knowledge and multi-step deduction rather than narrow math-only reasoning. We evaluate all methods with 5-shot chain-of-thought prompting and greedy decoding at temperature zero, matching the protocol reported for Qwen3 on MMLU-Pro. Accuracy is the fraction of questions whose final multiple-choice answer matches the label after answer extraction from the model output.

- **MultiPL-E** (Cassano et al., 2023): A polyglot extension of function-level code benchmarks (derived from HumanEval and MBPP) into many programming languages via parallel problem statements. Each item provides a natural-language specification and asks the model to complete an executable program; solutions are checked with language-specific unit tests. We use the standard Python split and report **pass@1** with one sample per problem at temperature zero, consistent with our other code suites. MultiPL-E probes whether math-only RL preserves general program synthesis from docstring-style prompts.

- **LiveCodeBench v6** (Jain et al., 2025): A contamination-mitigated code benchmark built from periodically released competition problems (e.g., LeetCode, AtCoder, Codeforces) with explicit time windows so that training-data overlap is minimized. Problems require algorithmic reasoning beyond short API memorization. We report results on the **v6** release split used in our Qwen evaluation harness. As with MultiPL-E, we use pass@1 with a single greedy sample per problem; hidden tests determine correctness.

- **HumanEval** (Chen et al., 2021): 164 hand-authored Python function-completion tasks. Each problem supplies a function signature, docstring, and partial implementation; the model must generate the remaining body. Correctness is verified by running official unit tests in an isolated environment. HumanEval is widely used for open models trained with English-centric code corpora; we include it for the DeepSeek family to align with common open-source reporting. Metric: pass@1, temperature zero, one sample per task.

- **MBPP** (Austin et al., 2021): Mostly entry-level Python problems stated in short natural language (974 tasks in the full corpus; we evaluate on the standard test split used in open LLM leaderboards). Compared to HumanEval, MBPP emphasizes straightforward scripting and standard-library use. It complements HumanEval by covering a broader set of short programming patterns. Metric: pass@1, temperature zero, one sample per task.

**Shared evaluation protocol.** All methods in Table 2 share the pretrained MoE students, math-only DeepScaleR training data, RLVR reward design, and decoding budget described in Section 3.1 and Appendix A. Checkpoints are selected by the same rule as the main math experiments. For each random seed in $\{0, 1, 2\}$, we run all three suites for the corresponding family, then compute the unweighted arithmetic mean of the three benchmark accuracies (reported as "Avg" in Table 2). Main-text tables report means over seeds; we do not apply additional test-time scaling or ensembling on non-math tasks. Because optimization targets math verifiable rewards only, some regression on code suites relative to the untrained Base is expected even when knowledge (MMLU-Pro) remains stable.

## E. Complete Ablation Results

*Table 8.* Complete ablation study results on Qwen and DeepSeek families across all benchmarks.

| Method | Qwen Family | | | | | | DeepSeek Family | | | | | |
|---|---|---|---|---|---|---|---|---|---|---|---|---|
| | AIME24 | AMC23 | MATH500 | Minerva | Olymp. | Avg | AIME24 | AMC23 | MATH500 | Minerva | Olymp. | Avg |
| **PADD (Full)** | **83.0** | **95.9** | **96.4** | **55.0** | **70.7** | **80.2** | **57.6** | **69.5** | **59.3** | **39.8** | **49.7** | **55.2** |
| PADD w/o Stage I | 79.1 | 87.1 | 91.8 | 52.2 | 60.3 | 74.1 | 54.9 | 63.2 | 54.1 | 37.8 | 44.7 | 50.9 |
| PADD w/o Stage II | 79.2 | 90.7 | 92.4 | 52.3 | 60.8 | 75.1 | 55.2 | 65.8 | 57.1 | 37.9 | 44.3 | 52.1 |
| PADD w/o Stage III | 80.7 | 95.3 | 94.2 | 51.2 | 66.8 | 77.6 | 55.9 | 70.1 | 58.7 | 38.7 | 49.9 | 54.7 |
| PADD w/o Stage IV | 81.3 | 96.1 | 95.1 | 52.7 | 68.2 | 78.7 | 56.5 | 70.5 | 59.1 | 39.3 | 50.4 | 55.2 |

Table 8 reports complete ablation results on all five math benchmarks for both families; the **PADD (Full)** row matches **PADD (Ours)** in Table 1. Results are consistent with the analysis on three representative Qwen datasets in the main text (Figure 2). On Qwen family, removing Stage I causes average performance to drop from 80.2% to 74.1% (6.1-point drop), verifying the importance of expert differentiation initialization. Removing Stage II lowers the average to 75.1% (5.1-point drop), removing Stage III to 77.6% (2.6-point drop), and removing Stage IV to 78.7% (1.5-point drop). On DeepSeek family, stage contribution trends are consistent with Qwen family, but overall performance is relatively lower, consistent with DeepSeek family students' smaller capacity. Notably, all ablation variants meet baseline requirements: Stage II (w/o Stage II) outperforms Online KD on all datasets, and Stage III (w/o Stage III) outperforms RSPO and GSPO on all datasets, further verifying the necessity and effectiveness of PADD stage components.

**Stage IV on DeepSeek.** Table 8 shows that the DeepSeek full pipeline and the variant without Stage IV both average 55.2%. Stage IV therefore has little effect on the DeepSeek average in this setting. Per-task entries show positive contributions on AIME24 and Minerva and slight negative changes on AMC23 and OlympiadBench. This pattern is consistent with Stage IV helping mainly when residual expert-load imbalance remains after Stage III. On Qwen, removing Stage IV lowers the average from 80.2% to 78.7%, a gain of 1.5 percentage points when Stage IV is retained.

## F. Stage I Clustering Sensitivity

**Purpose.** Stage I depends on the activation subset used for clustering and on the internal cluster count before mapping to student experts. This appendix tests how sensitive the final math average is to these choices on Qwen.

We vary only Stage I inputs on Qwen3-30B-A3B with $N{=}128$ experts per layer and Top-$K{=}8$ routing, as in Appendix A. We change activation subset size $|\mathcal{D}_A|$ and internal K-Means cluster count $M$ before mapping or merging to $N$ student experts. Stages II–IV and evaluation otherwise match the main paper.

*Table 9.* Stage I sensitivity on Qwen (math Avg %; default 80.2%).

| Variant | $|\mathcal{D}_A|$ | $M$ | Qwen Avg (%) | $\Delta$ |
|---|---|---|---|---|
| Default | 10% | 128 ($M{=}N$) | 80.2 | — |
| Subset-5% | 5% | 128 | 79.3 | −0.9 |
| Subset-20% | 20% | 128 | 80.4 | +0.2 |
| Overcluster-90 | 10% | 192 | 80.1 | −0.1 |
| Overcluster-120 | 10% | 256 | 79.7 | −0.5 |

Varying $|\mathcal{D}_A|$ or moderate over-clustering ($M{=}1.5N$ with hierarchical merge to $N$) changes the average by at most 0.9 points, indicating that Stage I is not brittle to reasonable clustering choices.

## G. Stage I Design Ablations

**Purpose.** Stage I uses cardinality-constrained clustering and a strict split between clustering data $\mathcal{D}_A$ and warmup data $\mathcal{D}_B$. This appendix compares these design choices to natural alternatives and provides NMI calibration analyses under the Qwen experimental configuration in Appendix A.

**Cardinality-constrained vs. standard K-Means.** Standard K-Means without capacity constraints can assign very different numbers of teacher neurons to each cluster. Table 10 reports the coefficient of variation of cluster sizes, defined as the standard deviation of cluster cardinalities divided by their mean; larger values indicate more uneven clusters. Uneven clusters map to imbalanced expert initialization and lower final accuracy. Constrained clustering keeps cluster sizes balanced and yields higher accuracy.

*Table 10.* Stage I clustering method ablation (Qwen math Avg %).

| $|\mathcal{D}_A|$ | Method | Cluster size CV | Avg (%) |
|---|---|---|---|
| 10% | Constrained (Ours) | 0.05 | **80.2** |
| 10% | Standard K-Means | 1.42 | 79.3 |
| 5% | Constrained (Ours) | 0.06 | 79.3 |
| 5% | Standard K-Means | 1.85 | 77.6 |

**Data partitioning between $\mathcal{D}_A$ and $\mathcal{D}_B$.** If warmup reuses the same samples that defined the cluster centroids, experts can memorize clustering-specific activations instead of generalizing.

*Table 11.* Effect of overlapping vs. partitioned warmup data (Qwen).

| Strategy | Warmup data | Train $L_{\text{KD}}$ | Val $L_{\text{KD}}$ | Avg (%) |
|---|---|---|---|---|
| Partitioned (Ours) | $\mathcal{D}_B$ only | 1.35 | 1.42 | **80.2** |
| Overlapping | $\mathcal{D}_A \cup \mathcal{D}_B$ | 1.18 | 1.68 | 78.8 |

**NMI under high routing entropy.** With $N{=}128$ and Top-8 routing, routing entropy is high and absolute NMI in Table 3 is therefore small. We add a uniform-random routing baseline on the same evaluation set:

*Table 12.* Supplementary NMI comparison (not in main Table 3).

| Routing strategy | NMI | vs. random |
|---|---|---|
| Random routing | 0.004 | 1.0× |
| Vanilla-GRPO | 0.013 | 3.3× |
| PADD (Stage I) | **0.030** | **7.5×** |

## H. Limitations and Broader Context

**Purpose.** This appendix states the scope of the problem setting, evaluation, and practical extensions that are not the focus of the main experiments.

**Problem setting.** As stated in Section 1, PADD distills from a router-less dense teacher into a pretrained router-aware MoE student and is complementary to dense-to-MoE architecture-initialization methods (Sparse Upcycling (Komatsuzaki et al., 2023), MoEfication (Zhang et al., 2022)), rather than a substitute for them. The two pipelines can be composed sequentially but require aligned starting checkpoints for a controlled head-to-head comparison.

**Evaluation scope.** Main results emphasize mathematical reasoning; Section 3.3 reports general-knowledge and code benchmarks under the same math-only training recipe. Primary training remains math-focused.

**Teacher scale and domains.** We use domain-specialized 7B teachers; see Appendix A.3. Scaling to 34B or 70B teachers may require offline logit caching, progressive distillation, or finer over-clustering as in Appendix I. Extending to a new domain typically requires a domain-appropriate teacher and re-running Stage I clustering, which takes about two to four GPU-hours in our setup.

# I. Clustering Quality Diagnosis and Robustification

Section 3.5 has described the structural prior, evaluation setup, and clustering quality monitoring overview. Here we supplement formal definitions of **diagnostic metrics** and operational details of **robustification procedures** for reproduction and extension.

## I.1. Diagnostic Metrics and Abbreviations

- **Silhouette (silhouette coefficient)**: Measures sample tightness within its cluster and separation from the nearest different cluster. For sample $i$, $s(i) = (b(i) - a(i)) / \max\{a(i), b(i)\}$, where $a(i)$ is the average distance from $i$ to other points in the same cluster, and $b(i)$ is the average distance from $i$ to the nearest different cluster. Silhouette closer to 1 indicates tight within-cluster and separated between-cluster, better quality.

- **Between-cluster variance**: Weighted squared deviation of cluster centroids from global centroid, reflecting between-cluster separability; appropriately high values indicate good cluster separation.

- **CV of cluster sizes**: Ratio of cluster sample count standard deviation to mean. Under capacity constraints, high CV indicates extremely uneven cluster sizes, which may affect routing and load balancing, requiring robustification.

## I.2. Method Summary and Robustification Procedures

We first obtain $N$ clusters using standard K-Means (or cardinality-constrained variant), then compute the above three metrics on a validation set; if Silhouette is low, between-cluster variance is too small, or CV is too large, we judge quality as poor. In this case, we adopt one of the following two strategies (or keep $N$ unchanged):

1. **Over-clustering with $M > N$ + hierarchical merging**: First perform K-Means with $M > N$ to get $M$ sub-clusters, then merge hierarchically (e.g., UPGMA, Unweighted Pair Group Method with Arithmetic Mean) by distance or similarity level by level until obtaining $N$ super-clusters; map super-clusters to student experts one-to-one, ensuring structural prior remains $N$-dimensional.

2. **GMM+BIC to select $M$ then align to $N$**: Fit neuron representations using Gaussian Mixture Model (GMM), select component number $M$ via BIC (Bayesian Information Criterion); if $M \neq N$, merge or split $M$ components (e.g., merge when $M > N$, subdivide using K-Means when $M < N$), finally obtaining $N$ equivalent clusters and mapping to student experts.

These procedures are completed in Stage I, do not change student MoE's expert number $N$, and do not participate in subsequent RL or distillation training.

# J. Expert Specialization Heatmaps and Structure Distillation

Section 3.5 has provided interpretation and conclusions for Figure 3. Here we supplement **heatmap construction** and statistical methods for reproduction. Each row corresponds to one Stage I-aligned cluster–expert index $j \in \{1, \ldots, N\}$ with $N{=}128$ on Qwen, as in Appendix A. Figure 3 in the main text plots a sampled subset of 60 experts for visualization.

Left figure is **teacher clusters$\times$categories**: After performing K-Means on teacher FFN $W_1$ to get $N$ clusters, we aggregate activation intensities of each cluster across different subdomains (algebra, geometry, etc.) on the validation set, forming a matrix by cluster $j$ and category $c$ and normalizing. Middle and right figures are **student experts$\times$categories**: On the same validation set, we compute the distribution of subdomains for tokens routed to each student expert $j$, obtaining expert $j$'s response intensity to category $c$; middle figure corresponds to model after Stage I only, right figure corresponds to model after Stages I–IV. All three heatmaps share the same columns (categories) and color scales, with row index $j$ corresponding by construction (teacher cluster $j \leftrightarrow$ student expert $j$), facilitating direct comparison. Subdomain labels are automatically generated from keywords and problem structure, unrelated to training or clustering.

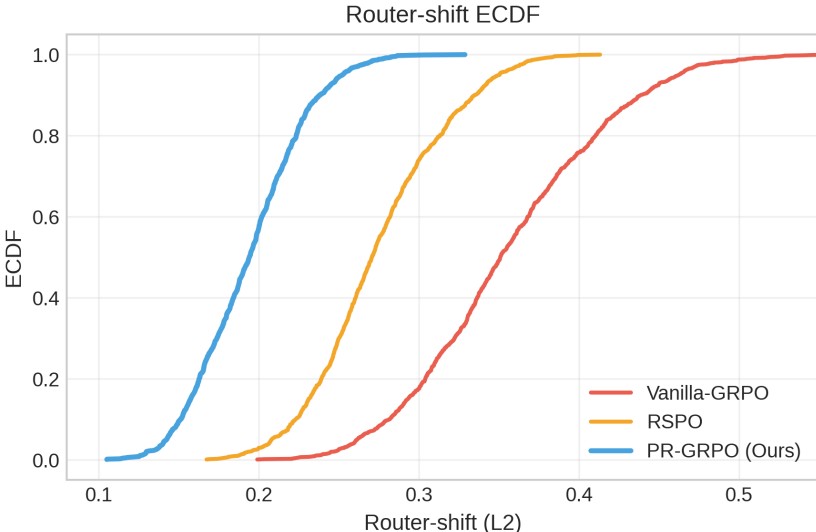

*Figure 5.* ECDF of Router-shift (aggregated near last steps). PR-GRPO shifts left with lighter tail, consistent with main text mean conclusion.

## K. Automated Subdomain Label Classification

To verify expert specialization effects, we need to classify math problems by subdomain. We use an automated classification method based on keyword matching and problem structure analysis, classifying problems into seven categories: algebra, geometry, number theory, probability, calculus, combinatorics, and others.

**Keyword matching.** First, we build keyword dictionaries for each subdomain. Algebra includes: equation, inequality, function, polynomial, root, coefficient, etc.; geometry includes: triangle, circle, area, volume, angle, similar, congruent, coordinate, etc.; number theory includes: prime, factor, divisibility, congruence, greatest common divisor, least common multiple, etc.; probability includes: probability, expectation, variance, combination, permutation, event, etc.; calculus includes: derivative, integral, limit, continuous, differentiable, etc.; combinatorics includes: permutation, combination, graph theory, counting, pigeonhole principle, etc.

**Problem structure analysis.** For problems that cannot be clearly classified via keywords, we analyze mathematical structure features. For example, problems involving coordinate systems and geometric transformations are classified as geometry; problems involving function properties and rates of change are classified as calculus; problems involving integer properties and divisibility relations are classified as number theory.

**Classification procedure.** For each problem, we first perform keyword matching; if keywords for a subdomain are matched with sufficient confidence, we classify directly; otherwise we perform structure analysis, classifying based on mathematical features; if still uncertain, we classify as "others." This classification method is used only for validation and visualization, not in training or initialization.

## L. Layer-wise Consistency and Distribution Evidence

Section 3.6 has provided global Router-shift curves, last three snapshot tables, and the conclusion that "layer-wise trends match the global curve." Here we supplement operational details and figures for **layer-wise** definitions and **distribution** evidence, without repeating main text conclusions.

### L.1. Layer-wise Router-shift

For each MoE layer $\ell$, we define $\Gamma^{(\ell)} = \|G_{\theta,\ell}(x_t) - G_{\theta_{\text{old}},\ell}(x_t)\|_2$ (isomorphic to Equation (7), restricted to layer $\ell$); evaluation setup is the same as the global figure (fixed batch, eval mode, routing noise disabled). We plot curves layer-wise and summarize mean for last three steps; all layers show the same ordering and magnitude as the global curve (Vanilla $\approx 0.34$–$0.36$, RSPO $\approx 0.26$–$0.28$, PR-GRPO $\approx 0.18$–$0.21$), with no opposite trends. Therefore, the main text's global

aggregation can be viewed as a robust summary of layer-wise stability improvements; layer-wise curves are provided in this appendix.

### L.2. Distribution Evidence: ECDF and Long Tail

Figure 4 in the main text only reports mean±CI. To test whether stability improvements are accompanied by distribution shape improvements (e.g., if PR-GRPO only lowers mean but has heavier tail, the proportion of high $\Gamma$ may not decrease), we compute **ECDF** on Router-shift samples at the same evaluation steps, showing ECDF curves for all three methods in the same figure. Figure 5 shows ECDF aggregated near the last steps: PR-GRPO consistently shifts left relative to RSPO and Vanilla with thinner right tail, meaning for any given threshold $\tau$ we have higher $\mathbb{P}(\Gamma \leq \tau)$ and lower proportion of high $\Gamma$, extending the main text's mean conclusion from a distribution perspective.

