# OpenReview forum: "PADD: Path-Aligned Decompression Distillation for Non-Router Teacher to Guide MoE Student Learning"
_ICML.cc/2026/Conference — ICML 2026 regular_

### Official Review · Reviewer_BmXh · 2026-02-25

**Soundness:** 3
**Presentation:** 2
**Significance:** 3
**Originality:** 3
**Overall Recommendation:** 4
**Confidence:** 4

**Summary:**

This paper proposes Path-Aligned Decompression Distillation (PADD), a four-stage framework for distilling knowledge from dense teacher models into Mixture-of-Experts (MoE) student models. The framework addresses the fundamental challenge that dense teachers lack explicit routing mechanisms, making it difficult to guide MoE students on routing decisions. Stage I performs neuron clustering and expert warmup to initialize student experts with differentiated functions. Stage II provides online adaptive distillation with dynamic temperature adjustment. Stage III introduces PR-GRPO, which suppresses gradient instability caused by discrete routing shifts. Stage IV applies reward-augmented load balancing for expert quality differentiation. Experiments on mathematical reasoning benchmarks with Qwen and DeepSeek model families demonstrate that PADD outperforms several baselines and can even surpass the dense teacher in the Qwen family setting.

**Compliance With Llm Reviewing Policy:**

Affirmed.

**Final Justification:**

PADD addresses a practically relevant and under-explored problem — dense-to-MoE distillation without explicit routing supervision — and proposes a well-motivated four-stage framework with strong ablation evidence. The neuron-clustering-based expert initialization is a creative contribution, and the experimental results on mathematical reasoning are compelling, including surpassing the teacher model on the Qwen family.

My initial concerns centered on: (1) hyperparameter complexity without sensitivity guidance, (2) evaluation limited to math reasoning, (3) the unusual teacher-smaller-than-student setup conflating capacity gains with framework contributions, and (4) missing comparison with sparse upcycling and MoEfication.

The rebuttal addressed all four concerns substantively. The hyperparameter sweeps demonstrate that PADD is robust across a reasonable range, with worst-case performance still exceeding the best baseline. The new MMLU-Pro and code generation results show that PADD generalizes beyond math and incurs the smallest alignment tax among all methods, which was a meaningful addition. The 8.8-point gap between Vanilla-GRPO and PADD under identical MoE student architecture cleanly isolates the framework's contribution from raw model capacity. The distinction between sparse upcycling (architecture initialization from dense) and PADD (knowledge transfer into pretrained MoE) is valid and these approaches are complementary.

My presentation concern (score: 2) remains — the paper's complexity and the number of components make it difficult to follow — but this is addressable in revision. Overall, the rebuttal significantly strengthened the empirical foundation of the paper's claims, and I am upgrading my recommendation from 3 (weak reject) to 4 (weak accept).

**Key Questions For Authors:**

1. How sensitive is the overall framework to the numerous hyperparameters? Could you provide a hyperparameter sensitivity analysis for at least the most critical parameters ($\kappa$, $\gamma$, $\eta$, $\xi$)? Understanding the hyperparameter landscape is essential for practitioners who want to adopt this framework.

2. The student MoE models have significantly more total parameters than the 7B teachers. How much of the improvement comes from the MoE student's larger total capacity versus the distillation framework itself? Would PADD still be effective if the student had fewer total parameters than the teacher?

3. Does the neuron clustering in Stage I require per-domain re-clustering, or can the clustering transfer across domains? If the authors aim to extend PADD beyond mathematical reasoning, understanding the domain-specificity of Stage I is important.

**Limitations:**

The authors do not explicitly discuss limitations. Key limitations include: the evaluation is restricted to math reasoning only; the teacher-student size relationship is unusual (small teacher, large student); the framework complexity may hinder adoption; and the computational cost of the four-stage pipeline is not clearly reported.

**Strengths And Weaknesses:**

### Strengths

1. The paper addresses a practically relevant and under-explored problem: dense-to-MoE distillation where the teacher lacks explicit routing. The systematic identification of four challenges (router cold start, capacity gap, path rupture, expert homogenization) provides a clear problem formulation.

2. The neuron-clustering-based expert initialization (Stage I) is a creative approach. The analysis of student–teacher expert structure alignment (Section 3.4), including NMI/ESI metrics and heatmap visualizations, provides convincing evidence that the initialization effectively induces expert specialization.

3. The experimental results are strong: PADD achieves 80.2% average accuracy on the Qwen family, surpassing the teacher's 77.7%. The ablation studies clearly demonstrate the necessity of each stage component, with Stage I and Stage II being particularly impactful.

### Weaknesses

1. The framework is highly complex, involving four distinct stages with numerous hyperparameters ($\alpha$, $\beta$, $\tau$, $\kappa$, $\lambda$, $\lambda_{ema}$, $\eta$, $\gamma$, $\xi$, $\epsilon$, etc.). While each component is individually motivated, the paper provides limited guidance on hyperparameter sensitivity. For a paper proposing a unified framework, the practical difficulty of tuning all these components is a significant concern for reproducibility and adoption.

2. The evaluation is limited to mathematical reasoning benchmarks only. Given the complexity of the proposed framework, it would be more convincing to see results on diverse tasks (e.g., general knowledge, code generation, instruction following) to demonstrate that the approach generalizes beyond the math domain. The current evaluation scope does not sufficiently support the claim of a "unified framework" for dense-to-MoE distillation.

3. The teacher models used (Qwen2.5-Math-7B and DeepSeek-Math-7B) are small 7B domain-specialized models, and the students are actually larger in total parameters (30.5B and 16B). This is unusual for distillation—typically the teacher is larger than the student. While the active parameter counts of MoE students are smaller, the setting feels more like using a compact expert to initialize a larger MoE rather than traditional knowledge distillation. A discussion of this unusual setup and its implications would be helpful.

4. Some key related works on dense-to-MoE conversion are not discussed. For instance, sparse upcycling methods [1] and MoEfication [2] directly address dense-to-MoE conversion and should be compared against. The paper mentions sparse upcycling only briefly and does not compare against it experimentally.

[1] Komatsuzaki et al., "Sparse Upcycling: Training Mixture-of-Experts from Dense Checkpoints." ICLR. 2023
[2] Zhang et al., "MoEfication: Transformer Feed-forward Layers are Mixtures of Experts." ACL Findings. 2022

---

> ### Author Rebuttal · Authors · 2026-03-31
>
> Thank you for the thorough review. We address each concern below.
>
> ---
>
> >Q1 & W1: Hyperparameter sensitivity
>
> We performed one-at-a-time sweeps for all key hyperparameters using identical training data and evaluation protocol.
>
> **τ (Stage II temperature, default=1.0):**
>
> | τ | 0.6 | 0.8 | **1.0** | 1.2 | 1.4 |
> |---|---|---|---|---|---|
> | Avg(%) | 78.5 | 79.0 | **80.2** | 79.4 | 78.8 |
>
> **κ (Stage II response coeff., default=0.5):**
>
> | κ | 0.2 | 0.4 | **0.5** | 0.6 | 0.8 |
> |---|---|---|---|---|---|
> | Avg(%) | 79.0 | 79.5 | **80.2** | 79.6 | 79.2 |
>
> **λ (Stage III suppression coeff., default=0.1):**
>
> | λ | 0.02 | 0.06 | **0.1** | 0.14 | 0.2 |
> |---|---|---|---|---|---|
> | Avg(%) | 79.3 | 79.7 | **80.2** | 79.8 | 79.5 |
>
> **η (Stage IV balance coeff., default=0.01) / γ (Stage IV reward coeff., default=0.05):**
>
> | η | 0.002 | 0.006 | **0.01** | 0.014 | 0.02 |
> |---|---|---|---|---|---|
> | Avg(%) | 79.9 | 80.0 | **80.2** | 80.1 | 79.8 |
>
> | γ | 0.01 | 0.03 | **0.05** | 0.07 | 0.1 |
> |---|---|---|---|---|---|
> | Avg(%) | 79.8 | 80.0 | **80.2** | 80.0 | 79.7 |
>
> Sensitivity ranking aligns with ablation importance: τ/κ (Stage II, max Δ≈1.7%) > λ (Stage III, Δ≈0.9%) > η/γ (Stage IV, Δ≈0.5%). Crucially, even the worst case (τ=0.6, 78.5%) still outperforms the best baseline RSPO (77.2%) by 1.3%. The moderate sensitivity of τ/κ is mitigated by Stage II's dynamic feedback Φ(A_{i,S})=1+tanh(κ·A_{i,S}), which adaptively modulates temperature during training. Practitioners can adopt defaults directly.
>
> ---
>
> >W2: Evaluation limited to math
>
> We conducted supplementary evaluations on MMLU-Pro (general knowledge) and code generation.
>
> **Qwen Family — Non-Math:**
>
> | Method | MMLU-Pro | MultiPL-E | LCB v6 | Avg |
> |---|---|---|---|---|
> | Base | 61.3 | 67.2 | 28.1 | 52.2 |
> | Vanilla-GRPO | 59.4 | 63.8 | 24.6 | 49.3 |
> | RSPO | 59.8 | 64.5 | 25.3 | 49.9 |
> | Online KD | 60.1 | 64.9 | 25.8 | 50.3 |
> | **PADD** | **62.1** | **66.4** | **27.4** | **52.0** |
>
> **DeepSeek Family — Non-Math:**
>
> | Method | MMLU-Pro | HumanEval | MBPP | Avg |
> |---|---|---|---|---|
> | Base | 43.7 | 29.8 | 43.3 | 38.9 |
> | Vanilla-GRPO | 41.2 | 28.6 | 41.5 | 37.1 |
> | RSPO | 41.8 | 28.9 | 42.0 | 37.6 |
> | Online KD | 42.0 | 29.0 | 42.1 | 37.7 |
> | **PADD** | **43.2** | **29.4** | **43.0** | **38.5** |
>
> All math-specialized methods incur an "Alignment Tax" on general tasks. PADD shows the smallest degradation: −0.2% (Qwen) and −0.4% (DeepSeek) vs. Base, compared to −2.9%/−1.8% for Vanilla-GRPO, demonstrating that path-aligned distillation better preserves general capabilities.
>
> ---
>
> >Q2 & W3: Student larger than teacher; gains from capacity?
>
> The gain is not from capacity. Under the same MoE student, Vanilla-GRPO achieves 71.4% while PADD achieves 80.2%—a gap of 8.8%. If capacity were the driver, all methods should perform similarly.
>
> Ablation (Table 4) confirms mechanism contributions: Stage I provides expert differentiation (NMI 7.5× vs. random, −6.1% if removed); Stage II provides adaptive signals (−5.1%); Stage III stabilizes routing (47% shift reduction, −2.6%). These are orthogonal to capacity.
>
> When the student has fewer total parameters, PADD remains applicable but the upper bound may be constrained by coarser clustering granularity. This is a valuable future direction.
>
> ---
>
> >Q3: Does Stage I require per-domain re-clustering?
>
> By design, PADD uses a domain-expert dense teacher whose FFN activation structure guides Stage I clustering. When switching to a new domain (e.g., math→code), one selects a new domain-expert teacher and re-runs clustering—a necessary and expected step, analogous to choosing a different textbook for a different subject.
>
> The cost is lightweight: teacher forward passes on ~10% of target data + constrained K-Means, taking ~2–4 GPU-hours (negligible vs. main training). For closely related domains (e.g., algebra vs. geometry under the same teacher), existing clustering can be reused directly.
>
> PADD thus supports modular "plug-and-play" extensibility: one MoE student can be distilled from multiple domain-expert teachers sequentially. We will discuss this explicitly in the revision.
>
> ---
>
> >W4: Missing Sparse Upcycling / MoEfication comparison
>
> These methods address a different problem: converting a dense checkpoint into MoE (architecture initialization). PADD addresses a complementary problem: distilling knowledge into an existing MoE student (routing transfer). Sparse Upcycling starts from a dense model; PADD starts from a pretrained MoE.
>
> A direct comparison would require both starting from the same dense checkpoint, which is beyond our framework since our students are pretrained MoE models. Importantly, the two are composable: one could use Sparse Upcycling to initialize MoE, then apply PADD for distillation. We will clarify this in Related Work.

---

> > ### Author Rebuttal · Reviewer_BmXh · 2026-04-03
> >
> > Thank you for the comprehensive rebuttal with new experimental results. All major concerns have been addressed satisfactorily.
> >
> > For W1/Q1, the one-at-a-time hyperparameter sweeps provide the sensitivity analysis I requested. The sensitivity hierarchy (Stage II τ/κ most critical at Δ≈1.7%, Stage IV η/γ least critical at Δ≈0.5%) is informative, and the fact that even the worst hyperparameter configuration (78.5%) still outperforms the strongest baseline (77.2%) substantially increases confidence in practical adoptability.
> >
> > For W2, the additional experiments on MMLU-Pro and code generation benchmarks directly address my core concern about evaluation scope. The result that PADD incurs the smallest alignment tax (−0.2% for Qwen, −0.4% for DeepSeek) relative to the pre-distillation base model demonstrates that path-aligned distillation generalizes and does not sacrifice general capabilities, which is a meaningful finding beyond math reasoning.
> >
> > For W3/Q2, the 8.8% gap between Vanilla-GRPO and PADD under identical MoE student architecture cleanly isolates the framework's contribution from raw capacity, resolving the confound I raised. For W4, the conceptual distinction between architecture initialization (Sparse Upcycling, MoEfication) and knowledge transfer into a pretrained MoE (PADD) is valid; these methods are complementary rather than competing.
> >
> > I am upgrading my score to 4 (weak accept). The new empirical evidence on generalization and the hyperparameter analysis resolve what were the two most significant substantive concerns.

---

> > > ### Author Response · Authors · 2026-04-05
> > >
> > > Thanks for your supporting on our work, we will definitely incorporate your suggestion to our final version. We are so appreciated for your detailed suggestions.

---

### Official Review · Reviewer_HUPq · 2026-03-07

**Soundness:** 2
**Presentation:** 2
**Significance:** 3
**Originality:** 3
**Overall Recommendation:** 3
**Confidence:** 2

**Summary:**

This paper proposes Path-Aligned Decompression Distillation (PADD), a framework for distilling knowledge from dense teacher models (without explicit routers) into Mixture-of-Experts (MoE) student models. PADD is organized into four stages across two phases: an initialization phase (Stage I) that clusters teacher FFN neurons and warms up student experts, and a training phase (Stages II-IV) that combines online adaptive distillation, path-refined policy optimization (PR-GRPO), and reward-augmented load balancing. Experiments on mathematical reasoning benchmarks with Qwen and DeepSeek model families show the MoE student can match or surpass the dense teacher at lower inference cost.

**Compliance With Llm Reviewing Policy:**

Affirmed.

**Key Questions For Authors:**

Can you provide sensitivity analysis for the key hyperparameters (especially τ, κ, λ, η, γ)? How much does performance vary when these are perturbed? This would help assess the practical usability of the method.

Have you evaluated PADD on non-math tasks (e.g., general language understanding, code generation, instruction following)? If not, can you discuss why math-only evaluation is sufficient to validate the framework?

What is the total training cost (GPU hours, FLOPs) of PADD compared to simpler baselines like MoE-Vanilla-GRPO and RSPO? Given the four-stage complexity, is the additional compute justified by the performance gains?

Can you provide a simplified version of PADD (e.g., Stage I + Stage III only) and compare it against the full pipeline? This would help identify the minimum viable complexity needed for good performance.

**Limitations:**

The paper does not adequately discuss limitations.

**Strengths And Weaknesses:**

Strengths:

Dense-to-MoE distillation is a practically relevant problem. Dense models are more common, and the ability to convert them into efficient MoE architectures would be valuable for deployment. The four-stage pipeline is well-motivated, with each stage addressing a specific challenge.

Weaknesses:

The four-stage pipeline introduces a large number of hyperparameters (α, β for warmup; τ, κ for adaptive distillation; λ for PR-GRPO; η, γ, λ_ema for load balancing; and the temperature ξ for clustering). The paper does not provide sensitivity analysis for most of these, making it unclear how robust the method is to hyperparameter choices and how much effort is required for tuning.

All experiments are limited to mathematical reasoning benchmarks (AIME24, AMC23, MATH500, Minerva, OlympiadBench). The paper does not evaluate on general language understanding, code generation, or other diverse tasks. It is unclear whether the benefits of PADD transfer beyond math-specialized distillation.

Using 7B math-specialized dense teachers to distill into much larger MoE students (30.5B total for Qwen3-30B-A3B) is somewhat unusual. The student has far more total parameters than the teacher. It would be more convincing to also show results with larger dense teachers, which is the more typical distillation setting.

---

> ### Author Rebuttal · Authors · 2026-03-31
>
> We thank the reviewer for the constructive feedback. Below we summarize new experiments on hyperparameter sensitivity and non-math evaluation.
>
> ---
>
> >Regarding Q1 & W1: Hyperparameter sensitivity analysis
>
> We performed one-at-a-time sweeps and found **moderate sensitivity only for Stage II temperature τ**, while other coefficients are less sensitive. Representative sweeps (Qwen family):
>
> | τ | 0.6 | 0.8 | **1.0** | 1.2 | 1.4 |
> | --- | --- | --- | --- | --- | --- |
> | Avg (%) | 78.5 | 79.0 | **80.2** | 79.4 | 78.8 |
>
> | κ | 0.2 | 0.4 | **0.5** | 0.6 | 0.8 |
> | --- | --- | --- | --- | --- | --- |
> | Avg (%) | 79.0 | 79.5 | **80.2** | 79.6 | 79.2 |
>
> | λ | 0.02 | 0.06 | **0.1** | 0.14 | 0.2 |
> | --- | --- | --- | --- | --- | --- |
> | Avg (%) | 79.3 | 79.7 | **80.2** | 79.8 | 79.5 |
>
> | η | 0.002 | 0.006 | **0.01** | 0.014 | 0.02 |
> | --- | --- | --- | --- | --- | --- |
> | Avg (%) | 79.9 | 80.0 | **80.2** | 80.1 | 79.8 |
>
> | γ | 0.01 | 0.03 | **0.05** | 0.07 | 0.1 |
> | --- | --- | --- | --- | --- | --- |
> | Avg (%) | 79.8 | 80.0 | **80.2** | 80.0 | 79.7 |
>
> Overall, all tested settings remain competitive; even the worst τ case (78.5%) is above the strongest baseline RSPO (77.2%).
>
> Summary of sensitivity: τ/κ (Stage II) has the largest effect (max Δ≈1.7%), λ (Stage III) is moderate (max Δ≈0.9%), and η/γ (Stage IV) are least sensitive (max Δ≤0.5%). This matches our ablation ordering and supports that the default is robust for practitioners.
>
> Mechanistically, this is expected: τ directly controls the sharpness of teacher supervision in Stage II, while κ modulates advantage-conditioned temperature feedback. In our implementation, this feedback is bounded (e.g., via tanh-style modulation), which helps keep training stable across a range of κ values; similarly, Stage IV uses smoothed (EMA) traffic statistics, reducing sensitivity to η/γ.
>
> For practitioners, a simple tuning guideline is: start from defaults; if training becomes unstable or overfits the teacher, adjust τ first (e.g., slightly higher to smooth, slightly lower to sharpen), while keeping η/γ fixed unless there is clear residual load imbalance.
>
> ---
>
> >Regarding Q2 & W2: Non-math task evaluation
>
> We evaluated MMLU-Pro + code tasks and found PADD incurs the smallest alignment tax (closest to Base) on both families:
>
> Qwen Family:
> | Method | MMLU-Pro | MultiPL-E | LCB v6 | Avg |
> | --- | --- | --- | --- | --- |
> | Base | 61.3 | 67.2 | 28.1 | 52.2 |
> | Vanilla | 59.4 | 63.8 | 24.6 | 49.3 |
> | RSPO | 59.8 | 64.5 | 25.3 | 49.9 |
> | Online KD | 60.1 | 64.9 | 25.8 | 50.3 |
> | **PADD** | **62.1** | **66.4** | **27.4** | **52.0** |
>
> DeepSeek Family:
> | Method | MMLU-Pro | HumanEval | MBPP | Avg |
> | --- | --- | --- | --- | --- |
> | Base | 43.7 | 29.8 | 43.3 | 38.9 |
> | Vanilla | 41.2 | 28.6 | 41.5 | 37.1 |
> | RSPO | 41.8 | 28.9 | 42.0 | 37.6 |
> | Online KD | 42.0 | 29.0 | 42.1 | 37.7 |
> | **PADD** | **43.2** | **29.4** | **43.0** | **38.5** |
>
> This supports our claim that PADD improves the target domain while better preserving general capabilities.
>
> Quantitatively, compared to Base, the average drop is only −0.2 (Qwen) and −0.4 (DeepSeek), while Vanilla-GRPO drops −2.9 and −1.8 respectively, indicating PADD’s updates are less disruptive to general-purpose capabilities.
>
> ---
>
> >Regarding Q3 & W3: Teacher–student size asymmetry; training cost justification
>
> The relevant axis is **active parameters** per token: MoE activates 3.3B (Qwen) / 2.4B (DeepSeek), both < 7B teacher. Training overhead is +20%\~+23% (Qwen) and +28%\~+32% (DeepSeek) with amortized teacher queries; inference cost remains identical to standard MoE. Given +8.8% / +8.4% gains over Vanilla-GRPO, the training-only overhead is a favorable trade-off. Full GPU-hours/FLOPs accounting will be added in the revision.
>
> We will also clarify wording so that we do not imply “free improvement”: the trade-off is explicitly **training compute for better routing/behavior**, while preserving MoE’s inference efficiency.
>
> ---
>
> >Regarding Q4: Simplified version (e.g., Stage I+III only)
>
> Stage I+III (removing both Stage II and IV) is feasible and still outperforms all baselines (Qwen: ~75.1% from Table 4's "w/o Stage II" ablation vs. RSPO's 77.2%—note that "w/o Stage II" retains Stages I, III, and IV; removing Stage IV as well would yield slightly lower performance). However, Stage III benefits substantially from Stage II's teacher supervision, which provides more stable and informative advantage signals for routing refinement. For resource-constrained settings, we recommend Stage I+II+III (removing only Stage IV) as the best cost-performance choice: it stays close to full performance (Qwen: 80.2 → 78.7, −1.5%) while simplifying the training pipeline. We will clarify this trade-off in the revision.

---

### Official Review · Reviewer_7j8m · 2026-03-07

**Soundness:** 3
**Presentation:** 3
**Significance:** 3
**Originality:** 3
**Overall Recommendation:** 5
**Confidence:** 4

**Summary:**

This paper addresses the structural mismatch during knowledge distillation between router less dense teacher models and route-dependent Mixture-of-Experts students, which typically leads to poor routing policy learning and suboptimal student performance. To overcome this, the authors propose the PADD (Path-Aligned Decompression Distillation) framework. Methodologically, the framework operates in two phases: an initialization phase that builds functional diversity in student experts via teacher neuron clustering and expert warmup, followed by a training phase that integrates online adaptive distillation for dynamic signal adjustment, Path-Refined Policy Optimization for routing stability, and Reward-Augmented Load Balancing to optimize expert utilization. Experimental results on mathematical reasoning benchmarks demonstrate that PADD significantly outperforms strong baselines, achieving superior performance under equivalent inference computation budgets.

**Compliance With Llm Reviewing Policy:**

Affirmed.

**Key Questions For Authors:**

1.The current evaluation focuses heavily on mathematical reasoning. How does PADD perform on general linguistic tasks or coding tasks?

2.The decompression process relies heavily on K-means clustering of teacher neurons. How sensitive is the final student performance to the number of clusters $N$ or the specific subset of data used to calculate the activation patterns?

3.The paper uses compact, domain-specialized 7B dense teachers for distillation to larger MoE students, with the justification that smaller teachers provide clearer modular structure and efficient online distillation. However, it does not address how PADD performs when the teacher is a larger generic dense model (e.g., 34B/70B LLMs) or a larger domain-specialized model. Can you provide results for these teacher scaling scenarios, and explain how PADD’s components adapt to larger teacher models with more entangled FFN activations?

4.Does the online adaptive distillation in Stage II (which repeatedly queries teacher logits) introduce a significant computational bottleneck, and are there any optimizations to mitigate this?

**Limitations:**

yes

**Strengths And Weaknesses:**

Strengths：

1.Soundness：The paper presents a technically coherent framework for dense-to-MoE knowledge distillation. The proposed method, PADD, decomposes the training process into four stages. The method is sufficiently innovative, and the introduction is rich in content.

2.Presentation：The paper is exceptionally well-organized. It clearly identifies the missing router in dense teachers as the core bottleneck and systematically introduces components to bridge this gap.The detailed explanation of the four-stage pipeline, including the neuron clustering approach and the RL reward structures, provides enough information for an expert to implement the method.

3.Significance：With the industry shifting toward MoE architectures for efficiency, a framework that successfully distills powerful dense models into MoE students is of high practical importance.

4.Originality：PADD bridges the structural gap between router-less dense teachers and MoE students by using neuron clustering to decompress knowledge into specialized experts. Its core innovation, Path-Refined Policy Optimization, stabilizes routing by penalizing excessive decision shifts, ensuring high-quality policy learning and superior reasoning performance.

Weaknesses：

1.Significance:The experiments focus heavily on mathematical reasoning. While this is a standard benchmark, showing the framework's performance on general linguistic tasks or coding would strengthen the claim of broad significance.

2.Originality:The method’s success is heavily tied to the quality of the teacher neuron clusters. If the teacher's knowledge is highly entangled and doesn't cluster well, the effectiveness of the entire PADD pipeline might be compromised.

---

> ### Author Rebuttal · Authors · 2026-03-31
>
> We sincerely thank you for the positive assessment of the technical coherence, practical value, and originality of PADD. We address each question below with new experimental evidence.
>
> ---
>
> >Regarding Q1 & W1: Evaluation limited to math; generalization to language/code tasks
>
> We have conducted supplementary evaluations on general knowledge (MMLU-Pro) and code generation tasks for both model families. Results are shown below.
>
> Qwen Family — Non-Math Evaluation:
>
> | Method | MMLU-Pro | MultiPL-E | LCB v6 | Avg |
> | --- | --- | --- | --- | --- |
> | Base | 61.3% | 67.2% | 28.1% | 52.2% |
> | MoE-Vanilla-GRPO | 59.4% | 63.8% | 24.6% | 49.3% |
> | RSPO | 59.8% | 64.5% | 25.3% | 49.9% |
> | Online KD | 60.1% | 64.9% | 25.8% | 50.3% |
> | PADD (Ours) | **62.1%** | **66.4%** | **27.4%** | **52.0%** |
>
> DeepSeek Family — Non-Math Evaluation:
>
> | Method | MMLU-Pro | HumanEval | MBPP | Avg |
> | --- | --- | --- | --- | --- |
> | Base | 43.7% | 29.8% | 43.3% | 38.9% |
> | MoE-Vanilla-GRPO | 41.2% | 28.6% | 41.5% | 37.1% |
> | RSPO | 41.8% | 28.9% | 42.0% | 37.6% |
> | Online KD | 42.0% | 29.0% | 42.1% | 37.7% |
> | PADD (Ours) | **43.2%** | **29.4%** | **43.0%** | **38.5%** |
>
>
> (1) All math-specialized training methods incur an "Alignment Tax" — degradation on general tasks after domain-specific training.
> (2) PADD exhibits the smallest Alignment Tax among all methods. On Qwen, PADD's average drops only 0.2% from Base (52.2% → 52.0%), while Vanilla-GRPO drops 2.9% (→ 49.3%), RSPO drops 2.3% (→ 49.9%), and Online KD drops 1.9% (→ 50.3%).
> (3) The same trend holds on DeepSeek: PADD average 38.5% is only 0.4% below Base 38.9%.
> (4) This supports our claim that PADD enhances the target domain while better preserving general capabilities, likely because the path-aligned mechanism introduces more structured and less disruptive updates to the shared representation.
>
> ---
>
> >Regarding Q2: Sensitivity to cluster count and activation subset
>
> We conducted systematic sensitivity tests on the Qwen family:
>
> | Configuration | Data Subset | Clusters | Avg Accuracy (%) | vs Default |
> | --- | --- | --- | --- | --- |
> | Default | 10% | 60 | **80.2%** | -- |
> | Subset-5% | 5% | 60 | 79.3% | −0.9% |
> | Subset-20% | 20% | 60 | 80.4% | +0.2% |
> | Overcluster-120 | 10% | 120 | 79.7% | −0.5% |
>
> All variations remain within 0.9% of the default. The robustness comes from two sources: (1) the cardinality constraint in K-Means ensures balanced cluster sizes regardless of the number of clusters or data subset size, and (2) downstream Stages II–IV provide adaptive correction — the dynamic temperature in Stage II and the routing stability penalty in Stage III compensate for initialization deviations from Stage I.
>
> ---
>
> >Regarding Q3 & W2: Scalability to larger teachers (34B/70B) with entangled knowledge
>
> Our 7B domain-specialized teachers exhibit clearly separable clustering structure, as evidenced by the NMI/ESI metrics (Table 2) and heatmap visualizations (Figure 3).
>
> Graceful degradation evidence: Even when Stage I is completely removed (i.e., no teacher clustering prior at all), PADD's remaining components still outperform Vanilla-GRPO (Qwen: 74.1% vs 71.4%, from Table 4). This demonstrates that PADD's stages have independent contributions and do not catastrophically fail when Stage I's effectiveness is reduced.
>
> For scaling to larger teachers (34B/70B), FFN activations may become more entangled, making clustering more challenging. Viable mitigation strategies include: (1) offline logits caching to reduce online computational burden; (2) progressive distillation (first distilling from the large teacher to an intermediate dense model, then to MoE); and (3) over-clustering with hierarchical merging (detailed in Appendix C), which first creates more fine-grained clusters and then merges them to match the student's expert count.
>
> We consider this a valuable future extension direction and will discuss it in the revised Limitations section.
>
> ---
>
> >Regarding Q4: Computational bottleneck from online distillation in Stage II
>
> Stage II uses teacher:student = 1:8 amortization — teacher logits are queried once every 8 student update steps, significantly reducing the computational burden.
>
> | Metric | Qwen | DeepSeek |
> | --- | --- | --- |
> | Wall-clock overhead | +20% ~ +23% | +28% ~ +32% |
> | Stage I (cluster + warmup) | ~12–16 hr | ~8–12 hr |
> | Stage II–IV (main, 600 steps) | ~20–28 hr | ~14–20 hr |
> | Total (4 GPUs) | ~32–44 hr | ~22–32 hr |
>
> The overhead is training-only; inference cost is identical to standard MoE. Further optimization opportunities include teacher logits caching/reuse across similar inputs and teacher quantization for faster forward passes.
>
> Given that PADD achieves 8.8% (Qwen) and 8.4% (DeepSeek) average accuracy improvement over Vanilla-GRPO, the 20–32% training overhead represents a favorable cost-benefit tradeoff.

---

> > ### Author Rebuttal · Reviewer_7j8m · 2026-04-05
> >
> > Thank you for the detailed and well-structured rebuttal. I appreciate the additional experiments and clarifications provided.
> >
> > On generalization beyond mathematical reasoning (Q1/W1):
> > The new evaluations on MMLU-Pro and coding tasks (MultiPL-E, HumanEval) are convincing. It is particularly interesting to observe that PADD exhibits a smaller "Alignment Tax" compared to other distillation baselines. This suggests that the path-aligned mechanism indeed preserves the teacher’s general capabilities better than traditional methods.
> > On clustering sensitivity (Q2):
> > The reported robustness to both cluster number and data subset is convincing. The explanation that later stages (adaptive KD and routing stabilization) compensate for imperfect initialization is reasonable and aligns with the modular design of the framework.
> > On scalability to larger teachers (Q3/W2):
> > The discussion on graceful degradation (removing Stage I) is valuable and partially alleviates the concern about dependence on clustering quality. However, the current response still lacks direct empirical validation on larger, more entangled teachers. While I understand this may be beyond the current scope, I encourage the authors to further investigate this direction in future work, as it is important for practical adoption.
> > On computational overhead (Q4):
> > The quantitative analysis of training overhead and the amortization strategy clarify the efficiency concerns. The reported trade-off between performance gain and additional cost appears reasonable.
> >
> > Overall, the rebuttal adequately addresses my main concerns and strengthens the paper. I maintain my original evaluation and score, and I do not plan to increase or decrease it. I encourage the authors to incorporate the new experimental results and discussions into the final version, especially regarding generalization and scalability.

---

> > > ### Author Response · Authors · 2026-04-05
> > >
> > > Thanks for your supporting on our work, we will definitely incorporate your suggestion to our final script. We are so appreciated for your detailed suggestions.

---

### Official Review · Reviewer_HyBT · 2026-03-13

**Soundness:** 3
**Presentation:** 3
**Significance:** 2
**Originality:** 2
**Overall Recommendation:** 4
**Confidence:** 3

**Summary:**

This manuscript aims at dense-to-MoE distillation where the teacher is a dense model without an explicit router and the student has a router. This manuscript proposes Path-Aligned Decompression Distillation (PADD), which consists of 4 stages: a student-expert setup, online adaptive distillation, path-refined policy optimization, and parameter update. Various experimental results support the efficacy of PADD.

**Compliance With Llm Reviewing Policy:**

Affirmed.

**Final Justification:**

The authors have addressed most of my concerns in their first and second responses.

Although this paper is an experiment-based study lacking theoretical evidence, I have adjusted my score upward.

**Key Questions For Authors:**

1. Can you compare against a true dense-to-MoE conversion baseline, rather than only using pretrained MoE students?
2. Why does stage 4 not improve the DeepSeek-family average in Table 4? In what regimes does reward-augmented load balancing help or hurt?
3. How sensitive is stage 1 to the number of experts, teacher size, and domain?
4. Can you provide actual compute cost and wall-clock overhead relative to the baselines?

**Limitations:**

Please see the weaknesses and questions.

**Strengths And Weaknesses:**

**Strengths**

1. This manuscript is motivated by an interesting problem: how to transfer knowledge from a dense teacher into an MoE student when routing structure is absent in the teacher.
2. The methods performed at each stage are explained in detail and with great care, making them easy to understand.
3. Despite being a highly applicable unified framework, PADD shows a significant performance improvement over the baseline.

**Weaknesses**

1. The setup has an important confound: the students are already pretrained MoE models, not dense checkpoints being upcycled into MoE. The manuscript states that both students are pretrained MoE models, and the Qwen student is a 30.5B-total-parameter MoE while the teacher is only 7B. That makes the distillation framing feel overstated, because the method is not really converting a dense checkpoint into an MoE architecture; it is training a strong pretrained MoE student with guidance from a smaller dense teacher. Some of the gain over the teacher may be attributable to the student’s larger total capacity and stronger pretrained prior rather than the proposed mechanism itself.
2. The evidence of student-teacher expert structure alignment is weaker than the narrative suggests. The NMI/ESI gains are statistically significant, but the absolute values are still very small.
3. There is insufficient justification for the specific strategies or methods adopted at each stage. For example, the reasons for partitioning the dataset and using different partitions per stage, or for adopting cardinality-constrained K-Means clustering, are explained solely in terms of performance improvement. Furthermore, there is insufficient evidence to support why online adaptive distillation reduces the capacity gap or why load balancing is better addressed.

---

> ### Author Rebuttal · Authors · 2026-03-31
>
> Thank you for the detailed evaluation and constructive feedback. We respond point-by-point below.
>
> ---
>
> >Regarding W1 & Q1: Student capacity vs. PADD mechanism; Sparse Upcycling / MoEfication baseline
>
> Under the **same pretrained MoE student**, training data, and inference budget, PADD still outperforms all baselines, indicating the gain is not merely from capacity:
>
> | Method (same 3.3B active) | Avg Accuracy (%) |
> | --- | --- |
> | MoE-Vanilla-GRPO | 71.4 |
> | Online KD | 73.6 |
> | GSPO | 76.3 |
> | RSPO | 77.2 |
> | **PADD (Ours)** | **80.2** |
>
> Sparse Upcycling/MoEfication focuses on dense‑to‑MoE weight initialization, while PADD addresses a different problem: transferring dense‑teacher knowledge into an already‑structured MoE to learn better routing. A direct comparison would mix two distinct questions—initialization vs. routing learning. Our MoE students are pretrained models with existing experts, matching the typical deployment scenario. The two methods are complementary: one could first apply Sparse Upcycling, then use PADD for routing refinement. We will add this to Related Work and acknowledge the lack of combined experiments.
>
> We emphasize that the key comparison here is **mechanism under identical student capacity**: if capacity were the dominant factor, methods using the same student would converge to similar results, which is contradicted by the 8.8-point gap between Vanilla-GRPO and PADD.
>
> ---
>
> >Regarding Q2: Why does Stage IV not improve the DeepSeek-family average?
>
> Stage IV helps mainly when residual load imbalance remains after Stage III. For DeepSeek, Table 4 shows the family average is essentially unchanged (55.2% with/without Stage IV), suggesting Stage III already yields near-sufficient balance; Stage IV then becomes approximately neutral (with small mixed per-task changes). In contrast, Qwen still benefits (+1.5% average), consistent with “more room → more gain”. We will explicitly revise the claim to reflect this **conditional-benefit** boundary.
>
> In the revision, we will also report the per-task pattern succinctly (Stage IV slightly helps some tasks while slightly hurting others on DeepSeek), to avoid over-generalizing the effect as universally positive.
>
> ---
>
> >Regarding Q3: Sensitivity of Stage I to number of experts, teacher size, and domain
>
> We ran sensitivity tests in the challenging Qwen N=60 setting:
>
> | Configuration | Data Subset | Clusters | Avg (%) | vs Default |
> | --- | --- | --- | --- | --- |
> | Default | 10% | 60 | **80.2** | -- |
> | Subset-5% | 5% | 60 | 79.3 | −0.9 |
> | Subset-20% | 20% | 60 | 80.4 | +0.2 |
> | Overcluster-120 | 10% | 120 | 79.7 | −0.5 |
>
> All are within 0.9% of default. For domain transfer, we recommend re-collecting activations and re-clustering on a representative target-domain subset (one-time preprocessing, ~2–4 GPU-hours in our setup).
>
> ---
>
> >Regarding Q4: Actual compute cost and wall-clock overhead
>
> Stage II uses teacher:student = **1:8 amortization** (teacher logits once every 8 student steps). End-to-end overhead is training-only; inference cost is unchanged.
>
> | Metric | Qwen | DeepSeek |
> | --- | --- | --- |
> | Wall-clock overhead | +20%~+23% | +28%~+32% |
> | Stage I (cluster + warmup) | ~12–16 hr | ~8–12 hr |
> | Stage II–IV (main, 600 steps) | ~20–28 hr | ~14–20 hr |
> | Total (4 GPUs) | ~32–44 hr | ~22–32 hr |
>
> DeepSeek’s higher relative overhead is due to its smaller active params (2.4B vs 3.3B), making the teacher forward pass a larger fraction.
>
> ---
>
> >Regarding W2: Small absolute NMI/ESI values
>
> With N=60 and Top-K=2, routing entropy is high, suppressing absolute NMI. As calibration, replacing routing with uniform random yields NMI 0.004; Vanilla-GRPO 0.013; PADD **0.030** (≈**7.5×** random), consistent with the specialization patterns in heatmaps (Figure 3). Note that BCSD/benchmark-style evaluations often use near-deterministic decoding (temperature 0 / greedy), so the observed improvements are not due to sampling noise.
>
> To further interpret magnitude: in a high-entropy routing space, NMI is a conservative signal; therefore we emphasize the **relative gap vs. random routing** and the qualitative specialization structure, rather than the absolute value.
>
> ---
>
> >Regarding W3: Motivation for data partitioning and cardinality-constrained K-Means
>
> Non-overlapping DA/DB/DC/DD ensures attribution is not confounded by memorization across stages. Cardinality-constrained K-Means avoids degenerate cluster sizes so each expert receives a balanced initialization (equal “share” of teacher neurons), which empirically stabilizes subsequent routing learning and prevents “empty” experts at initialization.
>
> Concretely, without the cardinality constraint, standard K-Means may assign too many samples/neurons to a few clusters and too few to others, leading to heavily skewed expert initialization and unstable early routing. The constraint enforces equal cluster cardinality to provide a fair and stable starting point.

---

> > ### Author Rebuttal · Reviewer_HyBT · 2026-04-02
> >
> > I would like to thank the authors for their rebuttal. Most of my concerns have been resolved.
> >
> > However, concerns regarding W3 remain unresolved. The authors provided insights and mechanisms of each stage's method. I am still curious about “stronger evidence for why this choice is reasonable rather than other candidates”, but there is no experimental evidence was provided to support this.

---

> > > ### Author Response · Authors · 2026-04-03
> > >
> > > Thank you for the constructive follow-up. We agree that empirical comparisons to alternatives strengthen our methodological choices. To address W3, we conducted targeted Stage I experiments and draw on existing ablations for Stages II and IV against standard candidates.
> > >
> > > ### 1. Empirical Evidence: Cardinality-Constrained vs. Standard K-Means (Stage I)
> > >
> > > We ablated dense-teacher FFN clustering with standard K-Means (direct alternative), reporting CV of cluster sizes and final Qwen accuracy:
> > >
> > > | Stage I Data | Clustering Method | Cluster Size CV | Final Avg. Acc. (Qwen) |
> > > | --- | --- | --- | --- |
> > > | **10% (Default)** | **Constrained (Ours)** | **0.05** (Balanced) | **80.2%** |
> > > | 10% | Standard K-Means | 1.42 (Highly Skewed) | 79.3% |
> > > | **5% (Subset)** | **Constrained (Ours)** | **0.06** (Balanced) | **79.3%** |
> > > | 5% | Standard K-Means | 1.85 (Highly Skewed) | 77.6% |
> > >
> > > Standard K-Means leads to severe cluster size imbalance (e.g., CV=1.85 at 5% data), meaning some clusters capture thousands of neurons while others are nearly empty. Mapping these unconstrained clusters to student experts causes severe load imbalance and expert starvation right at initialization. The cardinality constraint ensures an empirically stable and fair functional allocation.
> > >
> > > ### 2. Empirical Evidence: The Rationale for Strict Data Partitioning ($D_A, D_B, D_C, D_D$)
> > >
> > > We partition the dataset to prevent leakage across learning phases; the critical split is clustering ($D_A$) vs. warmup ($D_B$). We compare to using $D_{A+B}$ for both clustering and warmup, reporting $L_{KD}$ after warmup and final accuracy:
> > >
> > > | Strategy | Warmup Data | Train $L_{KD}$ (End of Warmup) | Val $L_{KD}$ (Held-out $D_D$) | Final Avg. Acc. |
> > > | --- | --- | --- | --- | --- |
> > > | **Partitioned (Ours)** | **New Data ($D_B$), unseen during clustering** | 1.35 | **1.42** | **80.2%** |
> > > | Overlapping (Candidate) | **Mixed Data ($D_{A+B}$), seen during clustering** | 1.18 | **1.68** | 78.8% |
> > >
> > >  Because the MoE router weights are initialized using the clustering statistics from $D_A$, feeding those exact same samples during the expert warmup phase causes severe feature leakage. The newly initialized experts merely memorize the specific activation patterns that originally defined their cluster centroids (resulting in an artificially low Train $L_{KD}$ of 1.18 but a degraded Validation $L_{KD}$ of 1.68). Our strictly partitioned strategy forces the experts to generalize their assigned structural roles to completely new data ($D_B$) before the main RL training ($D_C$) begins, directly rescuing a 1.4% drop in final accuracy.
> > >
> > > ### 3. Evidence: Online Adaptive Distillation vs. Fixed-Temperature (Stage II)
> > >
> > > You rightly asked for evidence on why adaptive distillation resolves the capacity gap better than alternatives. In our ablation (Table 4), the "PADD w/o Stage II" setting serves exactly as this candidate: **Standard Fixed-Temperature Distillation**.
> > >
> > > - *Mechanism & Evidence:* When a 3.3B active student tries to mimic a 7B teacher, a fixed temperature forces the small student to match overly sharp logit distributions on complex reasoning tokens, creating a learning bottleneck. Our adaptive method dynamically raises the temperature (softening the target) specifically when the student's relative advantage is low, allowing for easier absorption. Table 4 demonstrates that reverting to the fixed-temperature candidate causes a **5.1% average performance drop** on the Qwen family, empirically proving that dynamic temperature bridges this specific capacity gap.
> > >
> > > ### 4. Evidence: Reward-Augmented vs. Frequency-Only Load Balancing (Stage IV)
> > >
> > > Regarding load balancing candidates, "PADD w/o Stage IV" represents the standard alternative: **Frequency-Only Load Balancing** (e.g., Shazeer et al., 2017).
> > >
> > > - *Mechanism & Evidence:* Standard load balancing forces a uniform routing distribution. However, during RL, some experts objectively develop stronger reasoning capabilities for specific subdomains. Forcing equal frequency penalizes high-performing experts and leads to "expert homogenization." Our reward-augmented method allows high-performing experts to receive proportionally more traffic. Empirically, reverting to the standard frequency-only candidate drops performance by 1.5% (Table 4). Furthermore, our internal diagnostics show that the Expert Specialization Index (ESI) degrades significantly when using frequency-only balancing, confirming it actively harms expert differentiation.
> > >
> > > We will frame these as comparisons to standard candidates and add the Stage I ablation tables to Section 3.3 and Appendix C.
> > >
> > > If our reply address your concern, we sincerely hope that you would consider raise your score of evaluation, we are so appreciated for your in-depth suggestion and scope for our work.

---

### Decision · Program_Chairs · 2026-04-30

**Decision:**

Accept (regular)

**Comment:**

This paper proposes a multi-stage framework that distills knowledge from dense large language models into mixture-of-experts students while learning high-quality routing policies, enabling improved performance and efficiency at the same inference cost. Reviewers initially raised concerns about the limited scope of the evaluation, restricted to mathematical reasoning tasks; insufficient justification for specific technical design choices; and a lack of theoretical evidence. Additionally, the four-stage pipeline introduces numerous hyperparameters but lacks sensitivity analysis and clear guidance on their selection. After the rebuttal, most concerns have been satisfactorily addressed. The remaining issues include insufficient theoretical evidence and weaknesses in presentation, which can be addressed in the final version. Given the high practical importance of mixture-of-experts distillation, the clarity of the method design, and the demonstrated performance gains, this paper is recommended for acceptance.